# HERMES: Hybrid Error-corrector Model with inclusion of External Signals for nonstationary fashion time series

**Étienne David**                                      *etienne.david@heuritech.com*
*SAMOVAR, Télécom SudParis,*
*Institut Polytechnique de Paris, 91120 Palaiseau, France*

**Jean Bellot**                                        *jean.bellot@heuritech.com*
*Heuritech,*
*6 Rue de Braque, 75003 Paris*

**Sylvain Le Corff**                            *sylvain.le_corff@sorbonne-universite.fr*
*LPSM,*
*Sorbonne Université, UMR CNRS 8001, 75005, Paris*

**Reviewed on OpenReview:** *https://openreview.net/forum?id=4ofFo7D5GL&noteId=QnOhD911vc*

## Abstract

Developing models and algorithms to predict nonstationary time series is a long standing statistical problem. It is crucial for many applications, in particular for fashion or retail industries, to make optimal inventory decisions and avoid massive wastes. By tracking thousands of fashion trends on social media with state-of-the-art computer vision approaches, we propose a new model for fashion time series forecasting. Our contribution is twofold. We first provide publicly[1] a dataset gathering 10000 weekly fashion time series. As influence dynamics are the key of emerging trend detection, we associate with each time series an external weak signal representing behaviours of influencers. Secondly, to leverage such a dataset, we propose a new hybrid forecasting model[1]. Our approach combines per-time-series parametric models with seasonal components and a global recurrent neural network to include sporadic external signals. This hybrid model provides state-of-the-art results on the proposed fashion dataset, on the weekly time series of the M4 competition (Makridakis et al., 2018), and illustrates the benefit of the contribution of external weak signals.

## 1 Introduction

Multivariate time series forecasting is a widespread statistical problem with many applications, see for instance Särkkä (2013); Douc et al. (2014); Zucchini et al. (2017) and the numerous references therein. Parametric generative models provide explainable predictions with statistical guarantees owing to a precise modeling of the predictive distributions of new data based on a record of past observations. Calibrating these models, for instance using maximum likelihood inference, often requires a fair amount of tuning so as to design time-series-specific models able to provide accurate forecasts and sharp confidence intervals. Depending on the use case, statistical properties of the signal and the available data, many families of models have been proposed for time series. The exponential smoothing model (Brown & Meyer, 1961), the Trigonometric Box-Cox transform, ARMA errors, Trend, and Seasonal components model (TBATS) (Livera et al., 2011), or the ARIMA with the Box-Jenkins approach (Box et al., 2015) are for instance very popular parametric generative models. Hidden Markov models (HMM) are also widespread and presuppose that available observations are defined using missing data describing the dynamical system. This hidden state is assumed to be a Markov chain such that at each time step the received observation is a random

---

[1]https://github.com/etidav/HERMES

function of the corresponding latent data. Although hidden states are modeled as a Markov chain, the observations arising therefrom have a complex statistical structure. In various applications where signals exhibit non-stationarities such as trends and seasonality, classical HMM are not adapted. However, Touron (2017) recently proposed seasonal HMM, assuming that transition probabilities between the states, as well as the emission distributions, are not constant in time but evolve in a periodic manner. Strong consistency results were established in Touron (2019) and Expectation Maximization based numerical experiments were proposed. Although these works provide promising results, HMM are computationally expensive to train and are not yet well studied for seasonal sequences with thousands of components.

In many fields, single or few time series have become thousands of sequences with various statistical properties. In this new context, classical time-series-specific statistical models show limitations when dealing with numerous heterogeneous data. By constrast, recurrent neural networks and recent sequence to sequence deep learning architectures offer very appealing numerical alternatives thanks to their capability of leveraging any kind of heterogeneous multivariate data, see for instance Hochreiter & Schmidhuber (1997); Vaswani et al. (2017); Siami-Namini et al. (2018); Li et al. (2019); Lim et al. (2019); Salinas et al. (2020). The DeepAR model proposed in Salinas et al. (2020) provides a global model from many time series based on a multi-layer recurrent neural network with LSTM cells. More recently, applications using the Transformer model have been proposed (Li et al., 2019). The Temporal Fusion Transformers (TFT) approach is a direct alternative to the DeepAR model (Lim et al., 2019). Unfortunately, all these solutions suffer from two main weaknesses. Firstly, many of them are black-boxes as the final forecast usually does not come with a statistical guarantee although a few recent works focused on measuring uncertainty in recurrent neural networks, see Martin et al. (2021). Secondly, without a fine preprocessing and well chosen hyperparameters, these methods may lead to poor results and be outperformed by traditional statistical models, see Makridakis et al. (2018).

In this paper, we consider an emerging time series forecasting application referred to as *fashion trends prediction*. In fashion and retails industries, accurately anticipating consumers' needs is vital and wrong decisions can lead to massive wastes. With the explosion of social network and the recent advances in image recognition, it is possible to translate the visibility of fashion items on social media over time into time series. Consequently, models and algorithms can be trained to accurately anticipate and predict consumer behaviour. In Ma et al. (2020), a dataset is provided using social media pictures and an image recognition framework to detect several clothes: 2000 fashion time series are proposed with a weekly seasonality. However, only 3 years of historical data is available (144 data points) that may not be sufficient for some statistical approaches. In Ma et al. (2020), another dataset is presented gathering 8000 fashion sequences with an historical available data increased to 5 years. Nevertheless, only 120 values are available for each fashion time series and the overall volume remains low for a large part of the sequences resulting in a lot of noise and no clear patterns. In this paper, we propose a new fashion dataset overcoming the weaknesses of the two previous ones. Based on recent image recognition algorithms (Ren et al., 2015; Chollet, 2017), we built a large fashion dataset containing 10000 weekly sequences of fashion trends on social media with 5 years of historical data from 01-01-2015 to 30-12-2019. This dataset has very appealing properties: all time series have the same length (261 data points), there is no missing value and there is no sparse time series even for niche trends. Concerning fashion dynamics, some of them appear to be really volatile with nonlinear changes of dynamics resulting from the emergence of new tendencies. In this context, understanding early signals of the apparition of a trend is one of the key to accurately forecast the future of the fashion. Consequently, the originality of our dataset comes from the fact that additional external weak signals are introduced. With our fashion expertise, we detected several groups of highly influential fashion users. Analyzing their specific behaviours on social media, we associate with each time series an external weak signal representing the same fashion trends on this sub-category of users. They are called weak signals because they are often alerts or events that are too sparse, or too incomplete to allow on their own an accurate estimation of their impact on the prediction of the target signal. Exploring this new way of representing fashion, we aim at designing a model able to deal with such a large dataset, leveraging complex external weak signals and finally providing the most accurate forecasts.

Recurrent neural networks are appealing to tackle our forecasting problem due to their capability of leveraging external data. Recently, hybrid models combining deep neural network (DNN) architectures with widespread statistical models to deal with seasonality and trends have been proposed, see for instance Zhang (2003);

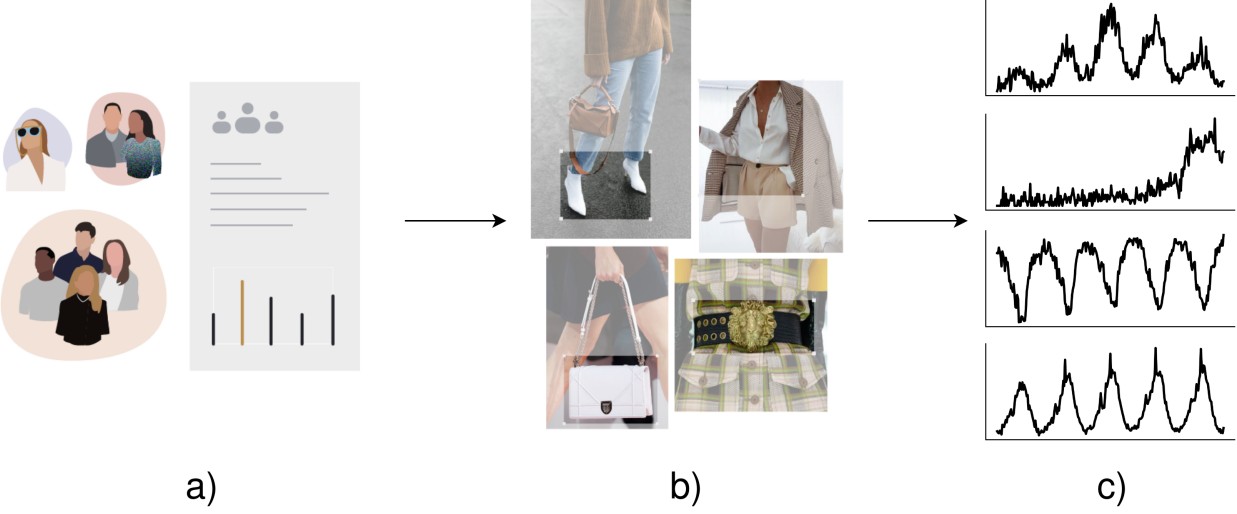

Figure 1: From social media to fashion time series. a) A complete image dataset of 150 millions of pictures is collected from social media users localized on 5 strategic markets. b) A visual recognition pipeline is applied on images. Global fashion items are detected with a collection of fine-grain attributes. c) Results are aggregated by fashion trend over time and normalized in order to remove social media bias.

Jianwei et al. (2019); Bandara et al. (2020). The approach providing the most striking results was proposed in Smyl (2020) in the context of the M4 forecasting competition (Makridakis et al., 2020). Given a large dataset, a per-time-series multiplicative exponential smoothing model was introduced to estimate simple but fundamental components for each time series and compute a first prediction. Then a global recurrent neural network was trained on the entire dataset to correct errors of the previous exponential smoothing models.

Following this work, we present in this paper HERMES, a new hybrid recurrent model for time series forecasting with inclusion of external signals. This new architecture is decomposed into two parts: First, a per-time-series parametric statistical model is trained on each sequence. Then, a global recurrent neural network is trained to evaluate and correct the forecast of the first collection of models. The ability to deal with external signals reveal the real potential of the hybrid approach: a global neural network, able to leverage large amounts of heterogeneous data, deal with any kind of external weak signals, learn context and finally correct weaknesses and errors of parametric models.

The paper is organized as follows. Section 2 presents the new fashion dataset provided with this article. Then, the proposed forecasting approach is presented in Section 3. Section 4 describes the experiments and comparisons with several benchmarks on 2 different use cases: the fashion dataset and the M4 competition weekly dataset. Finally, a general conclusion and some research perspectives are given in Section 5.

## 2 From social media to fashion time series

### 2.1 Translate fashion to data

An image dataset of 150 millions pictures is collected from different social media such as Instagram or Weibo. We targeted 5 strategic markets for the retail industry using posts localisation: the United States, Europe, Japan, Brazil and China. With compliance of privacy and data protection, only public accounts are selected and no potential private information is used, saved or revealed during the whole process. The second step consists in creating a powerful visual recognition framework to be able to detect clothes details on pictures like the type of clothing, the form, the size, the color, the texture, etc. To do so, the following framework is designed.

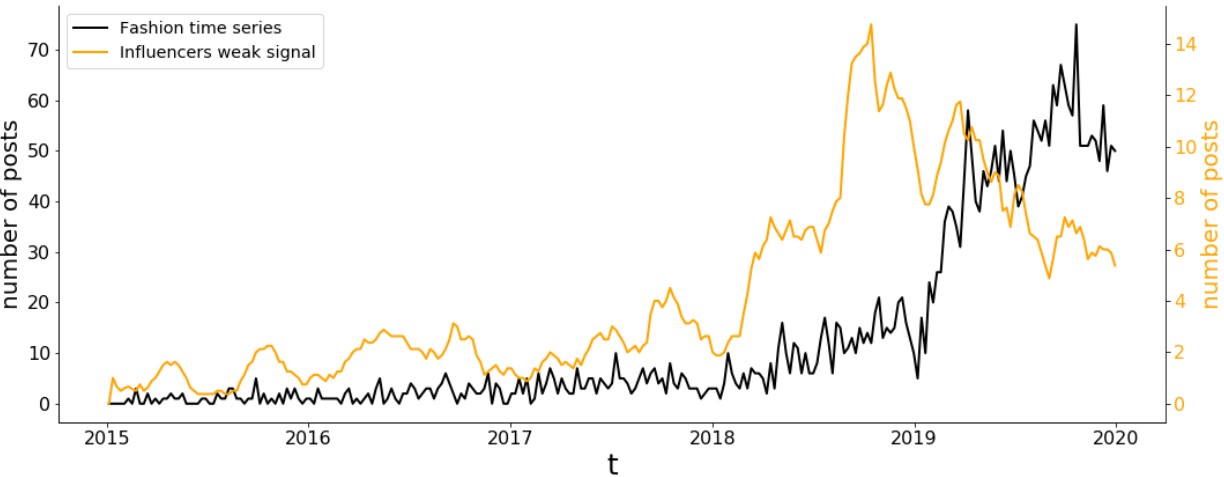

Figure 2: A shoes trend of the fashion dataset. In black the main signal and in orange its associated *fashion-forward* weak signal. The sudden explosion of the influencers signal at the end of 2018 announces the future burst of the trend in the mass market.

1. First, an object detection model is trained to detect the position, the size and the general type of possible multiple fashion items on a picture. This localization model is based on the Faster-RCNN architecture introduced in Ren et al. (2015). Starting from weights trained on MS-COCO (Lin et al., 2014), the model is fine-tuned with our data with a standard setup following the original paper.

2. Additionally, several visual recognition models are trained at classifying a rich collection of 350 fashion details. We train one classifier for each category of fashion item: one for pants, another for tops, a third for shoes, etc. These models are all based on the Xception architecture introduced in Chollet (2017). So as to trained them, large amount of social media pictures (between 200k and 800k training images depending on the category) have been manually tagged to constitute meaningful training datasets depending on the classification task. Architectures are first initialized with public weights trained on ImageNet (Russakovsky et al., 2014) and then fine tuned on the manually labeled dataset corresponding to their task.

At inference time, we first apply the localisation model which predicts boxes of generic fashion items (tops, pants, shoes, dresses, etc.) for each image. Then, each fashion item is cropped from its full image, resized to the classifiers' input size ($299 \times 299 \, \text{px}$) and fed into the related classifier: a top will be fed into the model trained on tops, etc. We obtain for each image a set of boxes, associated with a general category and a set of fine-grain attributes describing this object. As a final step, fashion experts aggregate those attributes to define relevant trends for the fashion and retails industry. We call them fashion trends. They can be just one attribute detected by a visual recognition model, for example the sneakers fashion trend. They can also be a combination of several attributes, for instance the Mini A-line dress fashion trend which is a combination of several attributes (dress length, shape, category,...) each one detected with a specific vision recognition model.

The 150 million of social media pictures are analyzed with this visual recognition pipeline. Out of those images, we detected clothes in 96 millions posts making the final dataset used in this paper. We aggregate results by fashion trend definition over the time and thousands of trends are finally translated from social media to time series. We note $y^{c,g,m,j}$ the final raw sequence representing the fashion trend $j$ of the cloth type $c$ for the gender $g$ on market $m$. At each time $t$, $y_t^{c,g,m,j}$ represents the number of posted pictures in the market $m$ during the week $t$ where computer vision algorithms detected the fashion trend $j$ of the cloth type $c$ for the gender $g$. As an illustration, example of fashion time series is given in Figure 2.

## 2.2 Removing social media bias

Due to the increasing use of social media and continuous changes of users' behaviours, a normalization step is applied to the raw sequences $y^{c,g,m,j}$ in order to remove bias. The pre-processing of data presented in this section does not constitute a technical contribution. This step is presented in depth to explain how the dataset provided with this paper is constructed. Let $\tilde{y}^{c,g,m}$ be the global sequence of the cloth type $c$ for the gender $g$ on market $m$ (e.g the evolution of the skirts in general for female in Europe). With the R package `stats`, the Seasonal-Trend decomposition using LOESS (Cleveland et al., 1990) is used to remove the seasonal component of $\tilde{y}^{c,g,m}$. The resulting deseasonalized signal is called $\bar{y}^{c,g,m}$. Finally, for any fashion trend $j$, the following normalized sequence is defined for all $0 \leq t \leq T$:

$$y_t^i = \frac{y_t^{c,g,m,j}}{\bar{y}_t^{c,g,m}} \,, \tag{1}$$

where T denotes the number of available time steps and $i$ represent a unique identifier summarizing the 4 information $c$, $g$, $m$ and $j$. The time series $y^{c,g,m,j}$ is divided by the deseasonalized signal $\bar{y}^{c,g,m}$ and not $\tilde{y}^{c,g,m}$ in order to avoid removing the seasonality of all the fashion trend sequences. With this normalizing step, most of the social media bias is removed and the final normalized sequences are expressed in share of category. As an illustration, an example of the normalization process is displayed in Figure 3. In this example, the raw and normalized time series of the Jersey top trend for females in China ($y^{c,g,m,j}$ with $c$=Top, $g$=Female, $m$=China and $j$=Jersey top) are presented. So as to compute the normalized signal, the raw time series of the Jersey top trend for females in China is divied by the deseasonalized raw time series representing the global Top trend for female in China ($\bar{y}^{c,g,m}$ with $c$=Top, $g$=Female and $m$=China).

## 2.3 Weak signal

In theoretical fashion dynamics (Rogers, 1962), different categories of adopters follow a trend in succession, resulting in several adoption waves. So as to catch the early signal of the emergence of a trend, 6000 social media influencers were selected by hand by fashion experts. Aggregating them, a specific "fashion-oriented" panel is created. With the same methodology as for the main panel described in Section 2.1 and Section 2.2, a normalized time series representing each fashion trend on this specific population is created. We named *fashion-forwards* this weak signal. For all fashion sequence $\{y_t^i\}_{1 \leq t \leq T}$, let $\{y_t^{f,i}\}_{1 \leq t \leq T}$ be the normalized sequence representing the behaviours of influencers regarding the fashion trend $i$. As we want to detect shifts between the main signal and the fashion forward signal, the following input is computed for the hybrid model: for all $t \in \{1, \ldots, T\}$ and any fashion trend $i$,

$$w_t^{f,i} = \frac{y_t^{f,i}}{y_t^{f,i} + y_t^i} \,.$$

where T denotes the number of available time steps. Values close to 0.5 indicate a similar behaviour between the influencers panel and the general panel. For instance, an emerging fashion shoes trend with its *fashion-forwards* weak signal is represented in Figure 2.

## 2.4 Fashion dataset

With this paper, we provide publicly[1] a sample of 10000 normalized fashion trends for men and women, over 9 different categories and 5 different markets. Each sequence has 261 time steps, from 2015-01-05 to 2019-12-31 with weekly values and no missing values. This collection of 10000 fashion trends was selected in order to represent finely the issues faced by the fashion industry. For instance, some sequences show complex behaviours with sudden changes, referred to as emerging or declining trends. A central point of this work is to accurately detect and forecast such trends. In addition, each fashion time series is linked with its associated normalized fashion forward signal as presented in the section above. An overview of the dataset can be found in Table 1. All the trends names are anonymized and for each trend, only the following macro information are revealed: the geolocalisation, the gender and the cloth type.

---

[1]`https://github.com/etidav/HERMES`

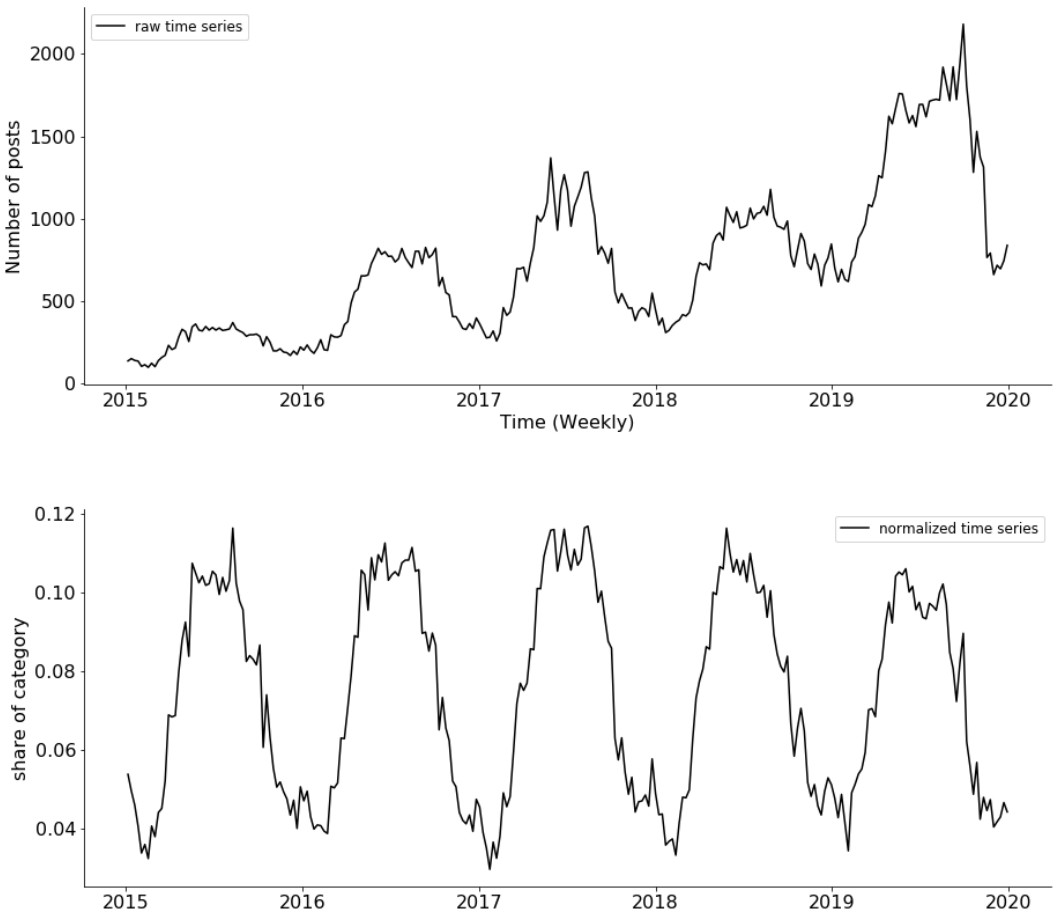

Figure 3: Example of difference between the raw sequence and the normalized one for the Jersey top fashion trend for females in China. In this example, the Jersey top fashion raw time series for females in China is normalized by the deseasonalized global top raw time series for females in China. The final normalized time series is finally expresssed as a share of the global catagery. (Top) Time series representing the raw signal of the Jersey top fashion trend for females in China. (Bottom) Time series representing the normalized signal of the Jersey top fashion trend for females in China.

Table 1: Fashion time series overview. For each couple geozone/category, the table gives the number of trends (Female/Male).

|  | Top | Pants | Short | Skirt | Dress | Coat | Shoes | Color | Texture |
|---|---|---|---|---|---|---|---|---|---|
| **United States** | 411/208 | 149/112 | 47/22 | 29/- | 20/- | 208/151 | 293/86 | 38/44 | 85/81 |
| **Europe** | 409/228 | 134/114 | 48/21 | 28/- | 20/- | 211/159 | 303/78 | 41/42 | 87/74 |
| **Japan** | 403/218 | 136/107 | 49/31 | 28/- | 23/- | 185/149 | 311/78 | 46/42 | 92/65 |
| **China** | 424/202 | 147/114 | 46/29 | 27/- | 27/- | 178/161 | 310/78 | 41/47 | 88/77 |
| **Brazil** | 431/222 | 134/117 | 49/27 | 30/- | 28/- | 203/152 | 311/76 | 48/41 | 107/84 |
|  |  |  |  |  |  |  |  |  |  |
| **Total** | 2078/1078 | 700/564 | 239/130 | 142/- | 118/- | 985/772 | 1528/396 | 214/216 | 459/381 |

# 3  HERMES: a new hybrid model for time series forecasting

We introduce a new hybrid approach for time series forecasting composed of two parts: a collection of per-time-series parametric models, and a global error-corrector neural network train on all time series. Per-time-series parametric models are used in particular to learn local behaviours and to normalize sequences by removing trends and seasonality. Then, a recurrent neural network driven by the weak signals is trained to correct these per-time-series models.

Consider $N \geqslant 1$ time series. For all $1 \leqslant n \leqslant N$ and $1 \leqslant t \leqslant T$, let $y_t^n$ be the value of the $n$-th sequence at time $t$ and $\mathbf{y}^n = \{y_t^n\}_{1 \leqslant t \leqslant T}$ be all the values of this sequence. The objective of this paper is to propose a model to forecast all time series in a given time frame $h \in \mathbb{N}$, i.e. we aim at sampling $\{y_{T+1:T+h}^n\}_{1 \leqslant n \leqslant N}$ based on $\{y_{1:T}^n\}_{1 \leqslant n \leqslant N}$.

## 3.1  Per-time-series predictors

For all $1 \leqslant n \leqslant N$, we note $f^n(.; \theta_{predictor}^n)$ the $n$-th parametric model of the $n$-th sequence where $\theta_{predictor}^n$ are unknown parameters. Given the sequences $\{y_{1:T}^n\}_{1 \leqslant n \leqslant N}$ and the estimated parameters $\{\theta_{predictor}^n\}_{1 \leqslant n \leqslant N}$, the time-series-specific forecasts $\{\widehat{y}_{T+1:T+h|T}^{pred,n}\}_{1 \leqslant n \leqslant N}$ are, for all $n \in \{1, \ldots, N\}$, for all $i \in \{1, \ldots, h\}$,

$$\widehat{y}_{T+i|T}^{pred,n} = f^n(y_{1:T}^n; \theta_{predictor}^n)_i. \tag{2}$$

During the M4 competition, the hybrid model of Smyl (2020) was based on a multiplicative exponential smoothing model as the time-series-specific predictor. However, on sporadic time series, this choice leads to poor results and instability. In this paper, a more general framework able to deal with any kind of per-time-series models is provided. Thus, the choice of the parametric model can be adjusted depending on the nature of the time series. The only limitation is the computational time as we aim at forecasting thousands of time series simultaneously. For instance, hidden Markov models provide a very interesting framework but inference of such models requires computationally costly iterative procedures such as Expectation Maximization-based algorithms often combined with Monte Carlo estimates of unknown expectations. Choosing these approaches as per-time-series predictors would considerably slow the overall training process of the hybrid model.

In Section 4, we introduce three declinations of the hybrid framework using different per-time-series predictors to highlight the adaptability of our approach. The first one is based on an exponential smoothing as a reference similar to the baseline Smyl (2020), the second one uses Thetam as per-time-series predictor (Hyndman et al., 2020) and the last one uses a TBATS model (Livera et al., 2011).

For non stationary time series, changes of behaviours are not always predictable using the past of the sequence. In some cases, these changes depend on external variables not considered by univariate parametric models. The difficulty is that the exact influence of external variables on the main signal is mostly unknown. This motivates the introduction of a global RNN trained on all time series and able to consider and leverage external signals.

## 3.2  Error-corrector recurrent model

The second part of the model is a global RNN, trained on all the $N$ sequences to correct the weaknesses of the first per-time-series parametric models. This task requires a thorough data pre-processing as recurrent neural networks training is highly sensitive to the scale of the data and requires well-designed inputs.

Let $w \in \mathbb{N}$ be the window size, usually this window is proportional to the forecast horizon $w \propto h$. The RNN input is defined as the following normalized, deseasonalized and rescaled sequence $\mathbf{z}_T^n = \{z_{T-w+i|T}^n\}_{1 \leqslant i \leqslant w}$: for all $1 \leqslant n \leqslant N$ and $1 \leqslant i \leqslant w$,

$$z_{T-w+i|T}^{n,T} := \frac{y_{T-w+i}^n - \widehat{y}_{T+k|T}^{pred,n}}{\bar{y}_T^n}, \quad \bar{y}_T^n = \frac{1}{w} \sum_{i=1}^{w} y_{T-w+i}^n.$$

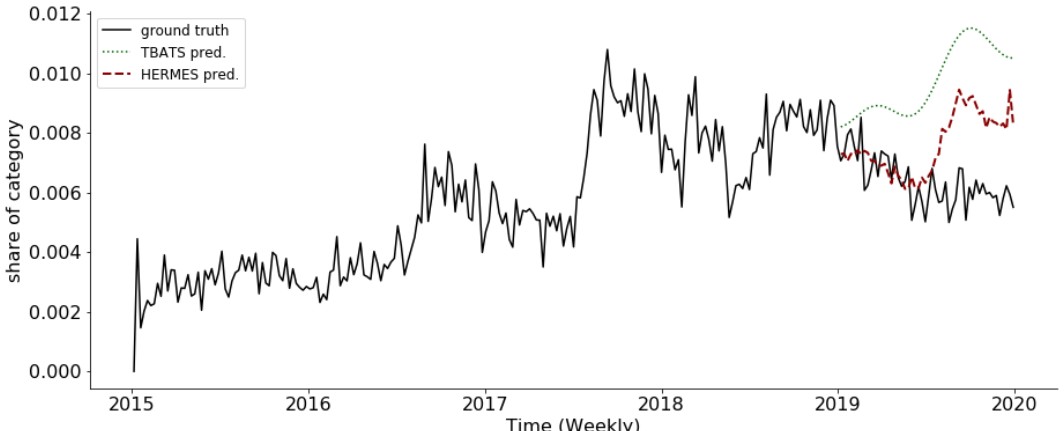

Figure 4: Hermes forecast example on a time series representing the vertical stipes texture fashion trend for females in Brazil. In green the prediction of the TBATS per-time-series predictors. In red the final forecast of our HERMES hybrid model.

where $k = i - h\lfloor i/h \rfloor$ with $\lfloor . \rfloor$ the floor function. With the numerator part $y^n_{T-w+i} - \widehat{y}^{pred,n}_{T+k|T}$, the per-time-series prediction is included in the RNN input and all the fundamental patterns already learned by this first predictor are removed from the time series. Then the denominator $\bar{y}^n_T$ is use to rescaled all input at the same level as the time series can have different scales. Another option could have been to divide directly $y^n_{T-w+i}$ by $\widehat{y}^{pred,n}_{T+k|T}$ but with time series hitting 0, this option is not valid. Let $\mathrm{RNN}(.; \theta_{corrector})$ be the recurrent neural network model where $\theta_{corrector}$ are unknown parameters. Given the RNN input sequences $\{\mathbf{z}^n_T\}_{1 \leqslant n \leqslant N}$ and the global RNN estimated parameters $\theta_{corrector}$, the correction terms $\{\widehat{y}^{corr,n}_{T+1:T+h|T}\}_{1 \leqslant n \leqslant N}$ are, for all $n \in \{1, \ldots, N\}$, for all $i \in \{1, \ldots, h\}$,

$$\widehat{y}^{corr,n}_{T+i|T} = \mathrm{RNN}(\mathbf{z}^n_T; \theta_{corrector})_i \cdot \bar{y}^n_T \ .$$

Thus, if no external signals are available, the final hermes forecast is, for all $1 \leqslant n \leqslant N$ and all $i \in \{1, \ldots, h\}$,

$$\begin{aligned}
\widehat{y}^n_{T+i|T} &= \widehat{y}^{pred,n}_{T+i|T} + \widehat{y}^{corr,n}_{T+i|T} \\
&= f^n(y^n_{1:T}; \theta^n_{predictor})_i + \mathrm{RNN}(\mathbf{z}^n_T; \theta_{corrector})_i \cdot \bar{y}^n_T .
\end{aligned} \tag{3}$$

### 3.3 Weak signal

In addition to the $N$ target time series, $K \times N$ external sequences indexed from 0 to $T$ are now considered. For all $1 \leqslant n \leqslant N$, $1 \leqslant k \leqslant K$ and $1 \leqslant t \leqslant T$, let $w^{n,k}_t$ be the value of the $k$-th external sequence at time $t$ associated with the sequence $\mathbf{y}^n$. Let $\mathbf{w}^n = \{\{w^{n,k}_t\}_{1 \leqslant t \leqslant T}\}_{1 \leqslant k \leqslant K}$ be all the values of the external signals. In addition, let $\mathbf{w}^n_T = \{\{w^{n,k}_{T-w+i}\}_{1 \leqslant i \leqslant w}\}_{1 \leqslant k \leqslant K}$ be only the last $w$ terms of the external sequences. Concatenating $\mathbf{z}^n_T$ and $\mathbf{w}^n_T$, a new input for the RNN is defined:

$$\mathbf{x}^n_T = \{x^n_{T-w+i|T}\}_{1 \leqslant i \leqslant w} = \{z^n_{T-w+i|T}, w^{n,1}_{T-w+i}, ..., w^{n,K}_{T-w+i}\}_{1 \leqslant i \leqslant w} \ .$$

Finally, for all $1 \leqslant n \leqslant N$ and for all $i \in \{1, \ldots, h\}$ the final prediction becomes:

$$\begin{aligned}
\widehat{y}^n_{T+i|T} &= \widehat{y}^{pred,n}_{T+i|T} + \widehat{y}^{corr,n}_{T+i|T} \\
&= f^n(y^n_{1:T}; \theta^n_{predictor})_i + \mathrm{RNN}(\mathbf{x}^n_T; \theta_{corrector})_i \cdot \bar{y}^n_T .
\end{aligned} \tag{4}$$

An illustration of the proposed model is displayed in Figure 5 and a first forecast example is given in Figure 4..

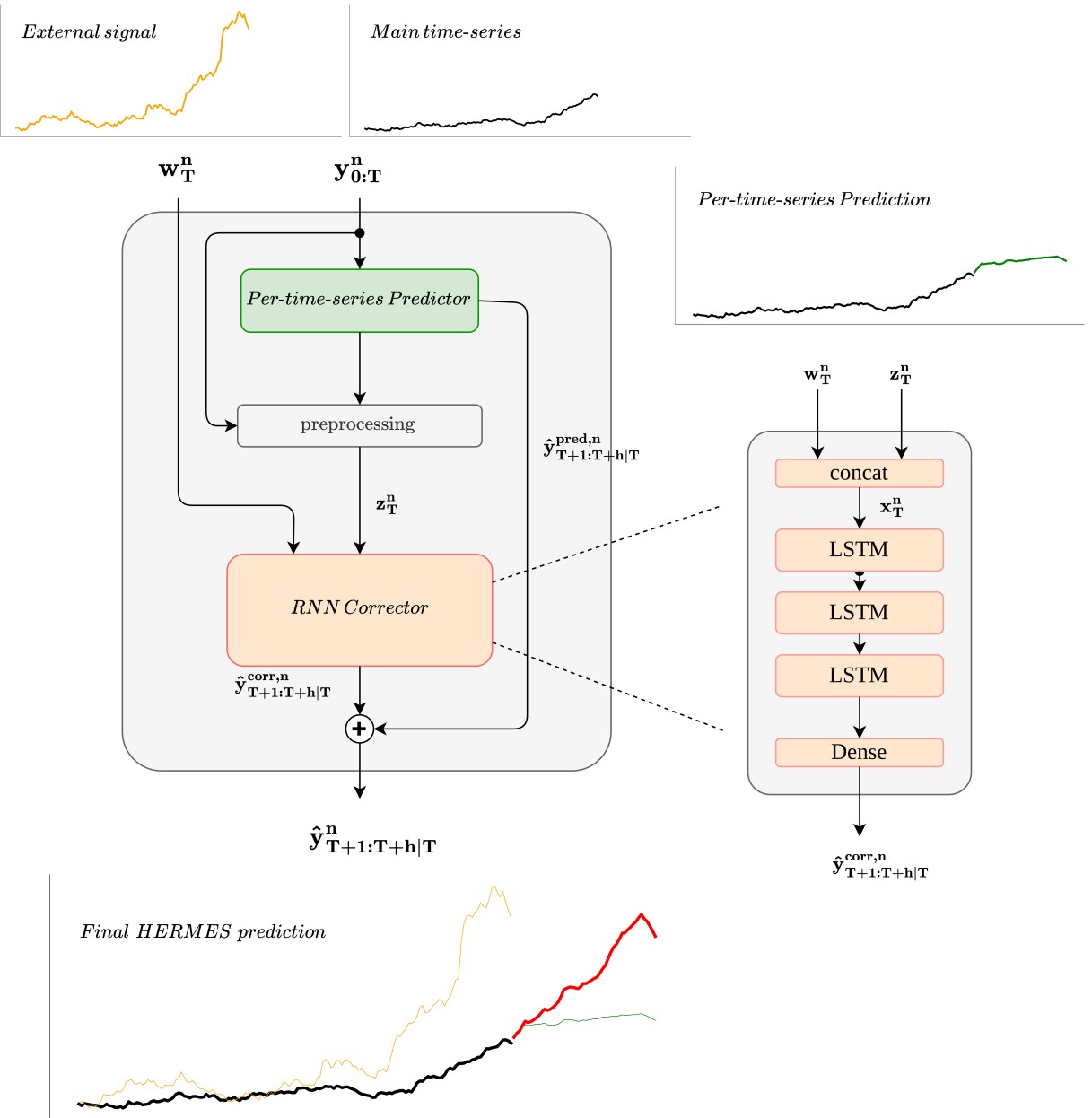

Figure 5: Architecture of the hybrid model with weak signals. The proposed framework can be decomposed in 5 steps: i) provide a time series. ii) (a) fit a first statistical model with the provided time series, (b) compute a first prediction and (c) preprocess the time series for the Global RNN. iii) If available, external signals can be added as part of the RNN input. iv) With a pre-trained RNN, compute a correction of the first statistical prediction. v) Compute the final forecast by adding the first time series prediction and the RNN correction.

## 4 Experimental results

In this section, performance of the hybrid model is assessed and compared with several other approaches. The HERMES framework is evaluated on two different use cases.

- A first application is proposed on the fashion dataset. The HERMES model is trained at forecasting the fashion time series one year ahead (h=52). This use case is mostly guided by the fashion and retail industry where clothes collections are usually prepared one year or more in advance. As additional signals representing influencers behaviours are available, this allows to set the focus on the ability of our framework to leverage these weak signals.

- A second application is proposed on the weekly dataset of the M4 competition (Makridakis et al., 2018). Based on the competition's rules, the forecasting horizon is set to 13 and no external signals are available. Furthermore, the M4 weekly time series come from different sectors and have variable lengths.

### 4.1 Fashion use case

#### 4.1.1 Training

The fashion dataset is split into three blocks, *train*, *eval* and *test* sets. The 3 first years are used as the *train* set, the 4th year is kept for the *eval* set and the *test* set is made of the last year. The hybrid model is trained to compute a one-year ahead prediction, $h$ equal to 52, and the window size $w$ is fixed at 104. Using the two first years of the *train* set, a first per-time-series parametric model for each time series is fitted. With the resulting collection of local models, a forecast of the third year is computed for each sequence. Corrector inputs are finally computed and the RNN is trained at correcting this first collection of third-year forecasts. For the *eval* set, per-time-series predictors are fitted a second time using the three first years and forecasts of the fourth year are computed. The *eval* set is used during training to control the learning of the RNN model and prevent overfitting. The per-time-series predictors are fitted a last time for the *test* set using the four first years. The final accuracy measures of all our models are computed on this *test* set. As an illustration, an example of our split is shown in Figure 6. Note that we have just enough historical data to perform the proposed train/eval/test split. Therefore, the common solution of applying a rolling window to increase the different splits cannot be used. Only one couple input/output is provided for each split per time series.

For the first parametric per-time-series models, existing Python libraries named `statsmodels` and `tbats` are used to estimate the different parameters $\theta^n_{predictor}$. The architecture of the recurrent neural network error-corrector model is composed of 3 LSTM layers of shape 50 and a final Dense layer to provide the correct output dimension. A classical Adam optimizer is used with a learning rate and a batch size set using a grid search. The loss function is defined as follows:

$$\ell(y^n_{T+1:T+h}, \widehat{y}^n_{T+1:T+h|T}) = \frac{1}{\bar{y}^n_T} \sum_{i=1}^{h} |y^n_{T+i} - \widehat{y}^n_{T+i|T}|.$$

This choice of $L_1$ loss function is motivated by its robustness to outliers which accounts for some time series in the fashion industry with very specific behaviours. The loss and previous parameters are all set with a complete grid search and have to be adapted regarding the use case. See C.1 for additional results concerning the loss function choice and C.2 for a complete grid search example. The code is developed in Python using the Tensorflow library and publicly available[1]. It allows the use of GPU to speed up the training process.

#### 4.1.2 Benchmarks, hybrid models and metrics

As benchmarks, several widespread statistical methods and deep learning approaches were selected. Using the R package `forecast` and the Python packages `statsmodels`, `tbats`, for each time series, predictions are computed with the following methods: *snaive*, *ets*, *stlm*, *thetam*, *tbats* and *auto.arima*. The forecast of

---

[1]`https://github.com/etidav/HERMES`

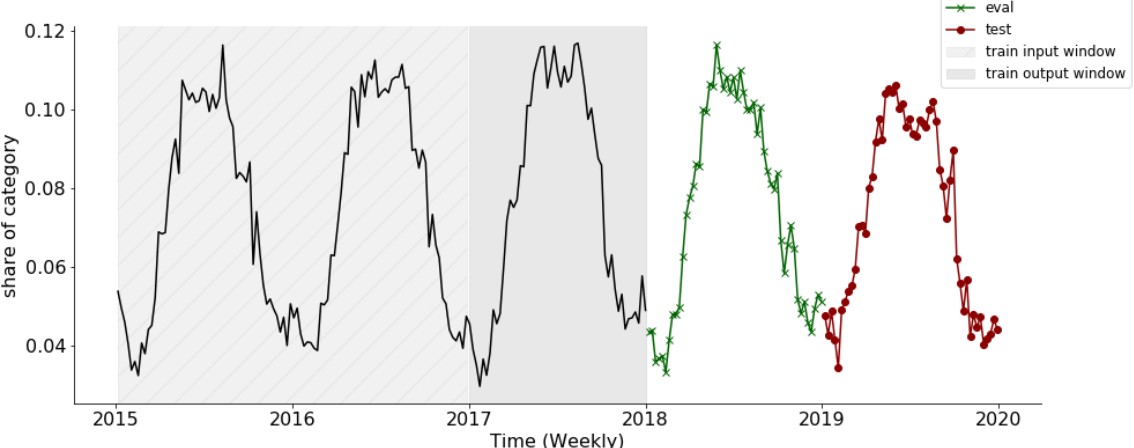

Figure 6: Temporal split for our training process. The three first years define our training set. The fourth year is used as our eval set and the final year is reserved for the test set.

the *snaive* method is only the repetition of the last past period. The *ets* model is an additive exponential smoothing with a level component and a seasonal component. The *stlm* approach uses a multiplicative decomposition and models the seasonally adjusted time series with an exponential smoothing model. The *Thetam* model decomposes the original signal in $\theta$-lines, predicts each one separately and recomposes them to produce the final forecast and *tbats* uses a trigonometrical seasonality. Finally, *auto.arima* is the R implementation of the ARIMA model with an automatic selection of the best parameters. A complete description and references for these models can be found in Hyndman et al. (2020).

We also compare our model with recent deep learning architectures for time series. The Prophet model introduced in Taylor & Letham (2017) is proposed and implemented with the available package bearing the same name `prophet`, and three models based on recurrent neural networks called *lstm*, *lstm-ws* and *deepar* are used. Concerning *lstm* and *lstm-ws*, they are both composed of 3 LSTM layers of shape 50 and a final Dense layer of shape 52. The first one (*lstm*) has only access to the main signal while the second one (*lstm-ws*) has access to the external signal. The last approach is the recent state-of-the-art model called DeepAR and introduced in Salinas et al. (2020). The Python package `GluonTS` (Alexandrov et al., 2020) was used to train DeepAR on the fashion dataset with a gridsearch to tune the main hyperparameters.

A strength of the proposed HERMES framework is that it can handle any kind of parametric model. Thus, three versions of HERMES are proposed on the fashion dataset using different local parametric models. The first one uses as predictors an additive exponential smoothing model as a reference close to Smyl (2020). The second one is based on the Thetam parametric model (Hyndman et al., 2020). Finally the last one relies on the TBATS model of Livera et al. (2011) and achieves the highest accuracy results on the fashion dataset. Declinations are called respectively *hermes-ets*, *hermes-tbats* and *hermes-thetam* according to the per-time-series model choice. For each of them, a variation with the inclusion of the weak signals (*-ws*) is presented.

To compare the different methods, we use the Mean Absolute Scaled Error (MASE) for seasonal time series. As our sequences have completely different scales, from $10^{-5}$ to $10^{-1}$, this metric was chosen to compute a fair error measure, independent of the scale of the sequence and suited for our seasonal fashion time series. The MASE metric is defined as follows, with $T$ the length of the time series, $m$ the seasonal period and $h$ the horizon:

$$\text{MASE} = \frac{T - m}{h} \frac{\sum_{j=1}^{h} |Y_{T+j} - \hat{Y}_{T+j}|}{\sum_{i=1}^{T-m} |Y_i - Y_{i-m}|} .$$

Table 2: Results summary on the 10000 time series of the fashion dataset. For each metric, the average on all our time series is computed. For approaches using neural networks, 10 models are trained with different seeds. The mean and the standard deviation of the 10 results are displayed. Models with a * in their name have access to the external signal.

| | MASE ↓ | | ACCURACY ↑ | |
|---|---|---|---|---|
| | *mean* | *std* | *mean* | *std* |
| *snaive* | 0.881 | - | 0.357 | - |
| *thetam* | 0.845 | - | 0.463 | - |
| *arima* | 0.826 | - | 0.464 | - |
| *ets* | 0.807 | - | 0.449 | - |
| *prophet* | 0.786 | - | 0.485 | - |
| *stlm* | 0.770 | - | 0.482 | - |
| *hermes-ets-ws\** | 0.769 | 0.005 | 0.501 | 0.007 |
| *hermes-thetam* | 0.764 | 0.003 | 0.497 | 0.005 |
| *hermes-thetam-ws\** | 0.760 | 0.004 | **0.520** | 0.010 |
| *hermes-ets* | 0.758 | 0.001 | 0.490 | 0.006 |
| *deepar* | 0.752 | 0.018 | 0.459 | 0.015 |
| *tbats* | 0.745 | - | 0.453 | - |
| *lstm-ws\** | 0.728 | 0.004 | 0.500 | 0.008 |
| *lstm* | 0.724 | 0.003 | 0.498 | 0.007 |
| *hermes-tbats* | 0.715 | 0.002 | 0.488 | 0.008 |
| ***hermes-tbats-ws\**** | **0.712** | 0.004 | 0.510 | 0.005 |

Detecting emerging and declining trends is a crucial issue for the fashion industry. A correct or incorrect prediction could lead to good returns or massive waste due to overstock or unsold clothes. In addition to the MASE accuracy metric, the different methods are also evaluated on a classification task and especially differences between methods using weak signals or not. In a given year, an increasing trend is defined as a trend that does more than 5% of growth on average with respect to the previous year. In the same way, a decreasing trend is defined as a trend that declines by 5% on average or more. Other trends are classified as flat trends. With this threshold, the proposed fashion dataset is almost balanced on the *test* set: There are 3087 increasing trends, 3342 decreasing trends and 3571 flat trends. To compare the different methods on this classification task, the accuracy metric, defined as the percentage of correct classification, is used.

### 4.1.3 Results

**10000 Fashion time series global accuracy.** For the two metrics and for each model, we compute the average on all sequences in the final year. Results are displayed in Table 2. For methods using neural networks, 10 models are trained with different seeds. The average and the standard deviation of their results are computed and displayed. For the statistical models, TBATS largely dominates the alternatives in terms of MASE. It is one of the main motivations why this model is used on the best HERMES candidate as the predictor model. All HERMES versions show consequent improvements regarding their per-time-series predictors in terms of MASE. *hermes-tbats* outperforms the two other declinations and all the benchmarks on the fashion dataset. This result highlights two important features of the proposed hybrid model: i) the final accuracy of the model is strongly impacted by the choice of the per-time-series predictor and ii) if the per-time-series predictor is well chosen regarding the nature of the time series, the resulting HERMES model seems to outperforms other state-of-the-art approaches.

Regarding the impact of the weak signals, Table 2 highlights an interesting improvement of the accuracy metric when weak signals are included. Figure 7 displays some examples of *hermes-tbats* predictions and illustrates some Tbats weaknesses that can be corrected by the hybrid approach.

**10000 Fashion time series classification task.** Classification results between the *tbats* model and the hybrid method *hermes-tbats* are given in Table 3, we note an impressive decrease of impactful errors: i.e.

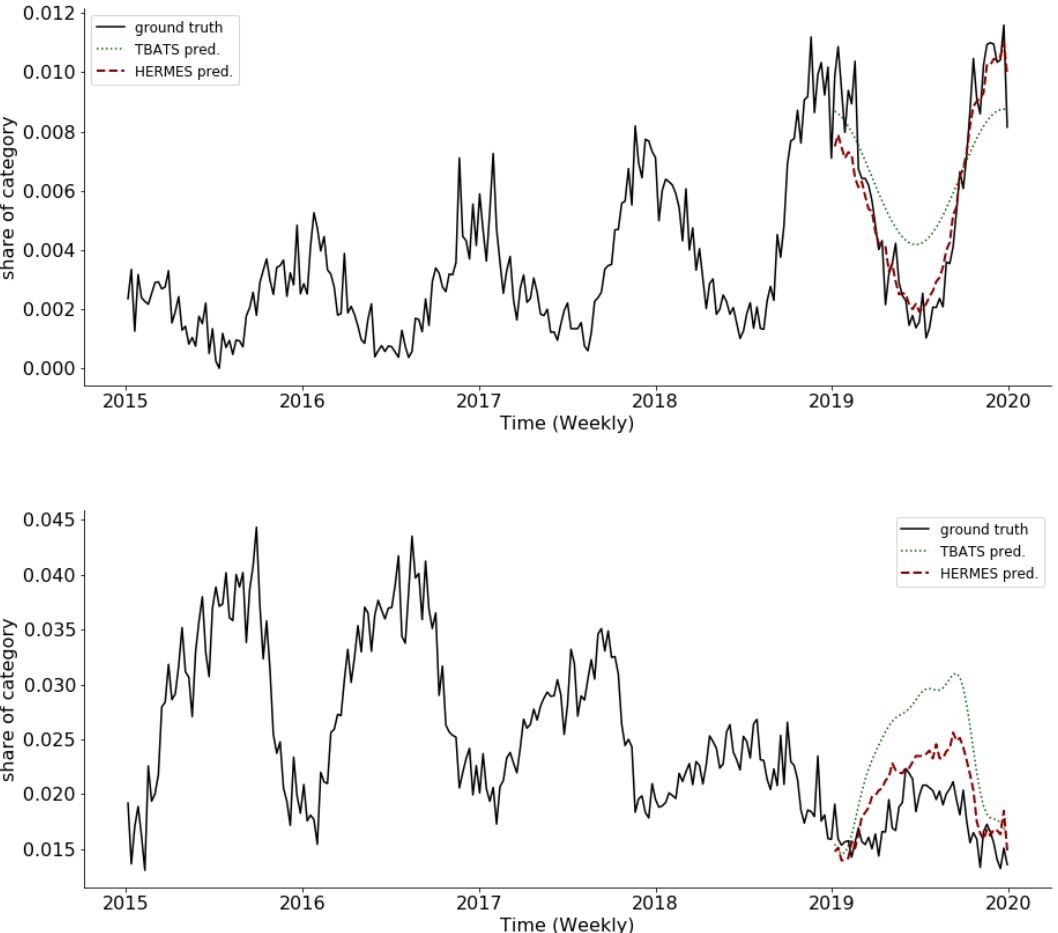

Figure 7: *hermes-tbats* forecast examples. In green the prediction of the per-time-series predictors *tbats*. In red the final forecast of our HERMES hybrid model *hermes-tbats*. (Top) Time series representing a top fashion trend for females in The United States. (Bottom) Time series representing the horizontal stipes texture fashion trend for females in China.

Table 3: *tbats*, *hermes-tbats* and *hermes-tbats-ws* models confusion matrix

| | **tbats** | | | | **hermes-tbats** | | |
|---|---|---|---|---|---|---|---|
| | pred-dec | pred-flat | pred-inc | | pred-dec | pred-flat | pred-inc |
| true-dec | 902 | 2113 | 327 | true-dec | 1261 | 1960 | 121 |
| true-flat | 351 | 2920 | 300 | true-flat | 549 | 2823 | 199 |
| true-inc | 300 | 2078 | 709 | true-inc | 214 | 2004 | 869 |

| | **hermes-tbats-ws** | | |
|---|---|---|---|
| | pred-dec | pred-flat | pred-inc |
| true-dec | 1956 | 1245 | 141 |
| true-flat | 1257 | 2087 | 227 |
| true-inc | 358 | 1620 | 1109 |

Table 4: Results summary on 1000 time series and 100 time series of the fashion dataset. The MASE average on all the time series is computed. For the two approaches using a neural network, 10 models with different seeds are trained. the mean and the standard deviation of the 10 results are displayed. Models with a * in their name have access to the external signal.

1000 time series fashion dataset

| | MASE ↓ | | ACCURACY ↑ | |
|---|---|---|---|---|
| | *mean* | *std* | *mean* | *std* |
| *snaive* | 0.871 | - | 0.383 | - |
| *thetam* | 0.837 | - | 0.484 | - |
| *arima* | 0.821 | - | 0.472 | - |
| *ets* | 0.801 | - | 0.469 | - |
| *prophet* | 0.788 | - | 0.476 | - |
| *hermes-ets* | 0.767 | 0.004 | 0.482 | 0.009 |
| *hermes-thetam* | 0.766 | 0.002 | 0.476 | 0.009 |
| *hermes-ets-ws** | 0.766 | 0.004 | **0.507** | 0.013 |
| *stlm* | 0.765 | - | 0.493 | - |
| *hermes-thetam-ws** | 0.763 | 0.005 | 0.501 | 0.009 |
| *lstm* | 0.740 | 0.007 | 0.487 | 0.014 |
| *deepar* | 0.738 | 0.017 | 0.465 | 0.013 |
| *tbats* | 0.734 | - | 0.466 | - |
| *lstm-ws** | 0.731 | 0.005 | 0.492 | 0.012 |
| *hermes-tbats* | 0.721 | 0.002 | 0.487 | 0.014 |
| ***hermes-tbats-ws*** | 0.717 | 0.004 | 0.500 | 0.010 |

100 time series fashion dataset

| | MASE ↓ | | ACCURACY ↑ | |
|---|---|---|---|---|
| | *mean* | *std* | *mean* | *std* |
| *snaive* | 0.876 | - | 0.330 | - |
| *thetam* | 0.822 | - | 0.470 | - |
| *arima* | 0.814 | - | 0.400 | - |
| *hermes-thetam* | 0.812 | 0.009 | 0.446 | 0.031 |
| *lstm* | 0.810 | 0.015 | 0.446 | 0.049 |
| *hermes-thetam-ws** | 0.810 | 0.008 | 0.479 | 0.024 |
| *deepar* | 0.804 | 0.024 | 0.393 | 0.029 |
| *hermes-ets-ws** | 0.792 | 0.003 | 0.386 | 0.010 |
| *hermes-ets* | 0.790 | 0.004 | 0.374 | 0.005 |
| *lstm-ws** | 0.789 | 0.010 | 0.485 | 0.036 |
| *ets* | 0.786 | - | 0.400 | - |
| *prophet* | 0.767 | - | **0.490** | - |
| *tbats* | 0.745 | - | 0.440 | - |
| *stlm* | 0.742 | - | 0.450 | - |
| *hermes-tbats* | 0.741 | 0.005 | 0.462 | 0.021 |
| ***hermes-tbats-ws*** | **0.737** | 0.004 | 0.486 | 0.027 |

forecasting an increase instead of a decrease and vice versa. The *hermes-tbats* model divides by 3 the error rate in comparison to *tbats* with only a slight decrease of the number of correct increase/decrease predictions. However, with our weak signals, we see that *hermes-tbats-ws* is able to catch twice as much as its relative model without weak signals while keeping a relatively low number of impactful errors.

**Size of the dataset.** In addition to the results on the whole fashion dataset, the robustness of the HERMES model is analyzed when it is trained on smaller datasets. Two experiments are performed on a sub sample of respectively 1000 and 100 randomly selected time series. Results are given in Table 4. The hybrid framework *hermes-tbats-ws* achieves the best performance in terms of global accuracy on both datasets. We can note that the accuracy of the full neural network *lstm* decreases when the dataset size decreases. On the small dataset of 100 time series, a local statistical model like *tbats* or *stlm* largely outperforms deep-model-based approaches such as *lstm* of *deepar*. In fact, providing sharp predictions from scratch is a complex task and high-dimensional recurrent neural networks require large amounts of data to do so. By contrast, the HERMES approach can rely on its first statistical part and consequently needs less data to be trained and to obtain interesting performance. We can nevertheless note that the gain brought by the error-corrector recurrent model decreases as the size of the dataset diminishes.

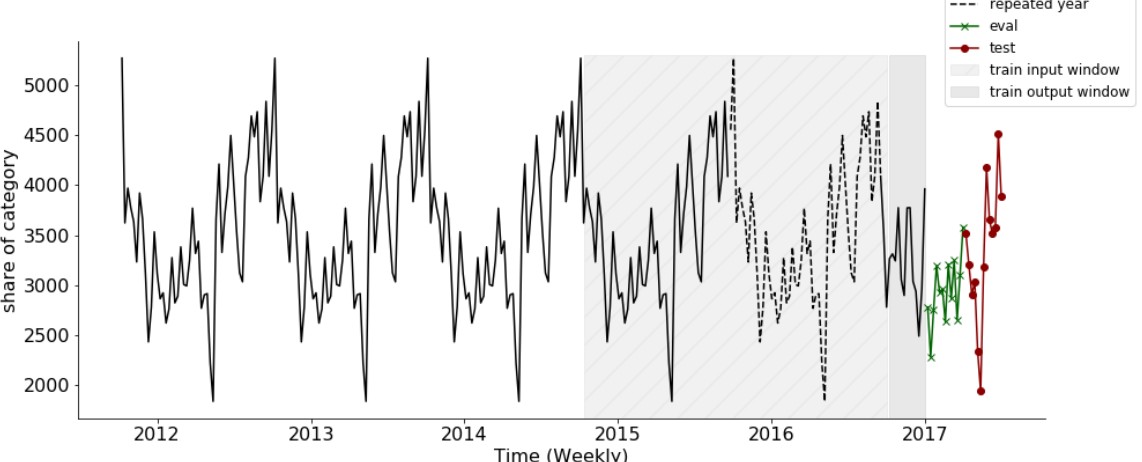

Figure 8: One of the shortest sequences of the M4 weekly dataset (93 time steps). In order to fit its predictor, the last complete year of the train set is duplicated in order to reach a total length of 300 time steps.

### 4.2  M4 competition use case

We also assessed the performance of HERMES using the M4 weekly dataset (Makridakis et al., 2020). The M4 dataset gathers 359 weekly time series and has 3 main differences compared to the proposed fashion dataset. Firstly, sequences do not have the same length with sequences lying between 93 and 2610 time steps. Secondly, the 359 time series come from different sectors such that finance or Industry. Accordingly, they have very distinct scales and dynamics. Thirdly, compared to the previous fashion application, the time horizon of the prediction is set to 13 for the weekly dataset and no additional external signals are provided.

#### 4.2.1  Training

The M4 dataset is preprocessed as follows. As some sequences are short (93 time steps), they limit the window size $w$ of the RNN. Consequently, 300 time steps are kept for each sequence. shorter sequences are duplicated in order to reach the length of 300 and longer sequences are cropped so as to keep the last 300 time steps. An overview of our train, eval, test set split and the resizing of the shortest sequences is given in Figure 8. Secondly, several M4 weekly time series have a large volume and a high level of variability. Consequently, Equation 2 of the HERMES framework is changed to:

$$\widehat{y}_{T+i|T}^{pred,n} = \exp\left(f^n(log\,(y_{1:T}^n)\,;\theta_{predictor}^n)_i\right)\,. \tag{5}$$

This simple modification increases significantly the accuracy of the per-time-series predictors tested on the M4 weekly dataset while reducing the fitting time. As for the fashion dataset, a complete grid search is done on the M4 weekly dataset to fix hyperparameters of the HERMES architecture. The horizon $h$ is set to 13 and the window size $w$ to 104. For the RNN part, the same architecture as described in Section 4.1.2 is used. The Adam optimizer is used and the MASE is directly used as the loss function. As some of the m4 weekly time series have many years of historical data, a rolling window is finally applied on the train set so as to increase the number of examples and improve the training process. The number of slinding windows, the learning rate, the batch size, the RNN architecture and input size are set using a grid search and detailled in C.2.

#### 4.2.2  Evaluation

The proposed model is evaluated along with a rich collection of benchmarks provided by the M4 competition, encompassing statistical models and neural network approaches. In addition, the hybrid model named *Uber* of S.Smyl is added. For a complete description and references of the benchmark models and the hybrid model

Table 5: Results summary on the m4 weekly dataset. For each metric, the average on all our time series is computed. For approaches using a neural network, 10 models are trained with different seeds. The mean and the standard deviation of the 10 results are displayed.

| | SMAPE | | MASE | | OWA | |
|---|---|---|---|---|---|---|
| | *mean* | *std* | *mean* | *std* | *mean* | *std* |
| *MLP* | 21.349 | - | 13.568 | - | 3.608 | - |
| *RNN* | 15.220 | - | 5.132 | - | 1.755 | - |
| *snaive* | 9.161 | - | 2.777 | - | 1.000 | - |
| *SES* | 9.012 | - | 2.685 | - | 0.975 | - |
| *Theta* | 9.093 | - | 2.637 | - | 0.971 | - |
| *Holt* | 9.708 | - | 2.420 | - | 0.966 | - |
| *Com* | 8.944 | - | 2.432 | - | 0.926 | - |
| *Damped* | 8.866 | - | 2.404 | - | 0.917 | - |
| *Uber* Smyl (2020) | 7.817 | - | 2.356 | - | 0.851 | - |
| *tbats* | 7.409 | - | 2.204 | - | 0.801 | - |
| ***hermes-tbats*** | **7.383** | 0.016 | **2.191** | 0.010 | **0.797** | 0.002 |
| *Pawlikowski, et al.* | 6.919 | - | 2.158 | - | 0.766 | - |
| *Petropoulos & Svetunkov* | 6.726 | - | 2.133 | - | 0.751 | - |
| *Darin & Stellwagen* | **6.582** | - | 2.107 | - | 0.739 | - |
| ***fforma-hermes*** | 6.614 | - | **2.058** | - | **0.732** | - |

*Uber*, see Makridakis et al. (2020) and Smyl (2020). As a HERMES candidate, a version using TBATS is proposed and called *hermes-tbats*. We propose a focus on the top 3 models reaching the highest accuracy on the M4 weekly dataset. These three methods are based on an ensembling and combine various approaches. The first model is presented in Darin & Stellwagen (2020) and called *Darin & Stellwagen*. The second model is introduced in Petropoulos & Svetunkov (2020) and called *Petropoulos & Svetunkov*. Finally, a description of the third model called *Pawlikowski, et al.* can be found in Pawlikowski & Chorowska (2020). An ensembling combining 4 HERMES variations is proposed. It is based on the FFORMA algorithm introduced in Montero-Manso et al. (2020) and called *fforma-hermes*. A complete description of the training process of the proposed ensembling is given in B.3. Following the M4 competition methodology, all the candidates are evaluated according to the MASE, the SMAPE and the OWA measures. A complete definition of these metrics is proposed in Makridakis et al. (2020) and summarized in B.1. See also B.1 for additional information about the M4 weekly dataset.

### 4.2.3 Results

The final results for the M4 weekly dataset are displayed in Table 5. The HERMES approach *hermes-tbats* outperforms all the benchmarks. This result is partially induced by the use of TBATS per-time-series predictors which achieve impressively good results on the test set. Regarding the hybrid model proposed by S.Smyl, its accuracy remains low in comparison to *tbats* and *hermes-tbats*. For the ensembling methods, the proposed FFORMA model with 4 HERMES variations *fforma-hermes* reaches the same high level of accuracy as the top 3 methods of the competition on the weekly dataset. The results provided by *hermes-tbats* confirm that the HERMES model is well suited for a large collection of forecasting tasks even difficult ones with small datasets, heterogeneous time series and the absence of additional useful external signals. Secondly, the accuracy gap between the proposed hybrid model and the approach proposed in Smyl (2020) illustrates the importance of a global framework able to leverage any kind of per-time-series predictors depending on the use cases. Finally, our model can be easily included as part of an ensembling method to improve the final robustness and accuracy of the predictions.

### 4.3 Discussion

Results of the two previous forecasting tasks illustrate that the proposed hybrid method is a general framework, easily adaptable to different use cases and able to compete with other state-of-the-art methods. With its neural network component, it is also able to leverage any kind of external signals, which which could become essential in more and more forecasting applications. The experiments highlight some limitations that may be the topic of future works.

- In its current form, the use of the neural network prevents the interpretability of the final prediction. The use of the external signal can not be easily assessed and no confidence intervals are provided with the hybrid framework. Future works can focus on designing more intepretable models, using for instance Bayesian neural networks, or combining neural networks with state space models.

- The proposed hybrid approach can be computationally intensive, for instance compared to classical neural-network-based models. A solution to improve this limitation is to propose new procedures to train jointly both parts of the model with mini batches of observations.

- In the Fashion use case, the inclusion and use of the external signals could be improved as the external signals are relevant only in a few fashion time series. An ongoing work is to use several neural networks specialized on different uses of the external signals. Depending on the past of the time series and the external signals, a hidden state could be used to switch between models.

## 5 Conclusion

In this paper, we propose a new hybrid model for non stationary time series forecasting. By mixing the performance of local parametric models and a global neural network, *hermes-tbats* clearly outperforms traditional statistical methods and full neural network models on two forecasting tasks. Furthermore, this new model is totally suited to deal with external signals. With a fine pre-processing and a well-designed architecture, the proposed hybrid framework succeeds at leveraging complex extra data and reaches promising accuracy levels. In addition, a fashion dataset gathering a sample of 10000 time series and a collection of weak signals is provided. We show that it is possible to represent fashion data with time series and introduce statistical models based on social media data. By making it publicly available, we hope that it will enhance the diversity of datasets for time series forecasting and pave the way for further explorations. As a possible future work, designing new models for the weak signals would improve their inclusion in the HERMES architecture. Focusing on the examples with important changes of behaviours, a fine analysis of the impact of the collection of weak signals is the topic of ongoing works. In the same way, an interesting improvement of the hybrid framework can be to introduce not a single but several neural networks trained at correcting different kinds of weaknesses. A perspective is to add a latent discrete label to select dynamically the regime shifts.

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

## A   Fashion dataset: a study of sub samples of time series sharing the same behaviour

A study of 4 subsamples of the fashion dataset is proposed in this section. They are defined as follows.

- **disrupting time series**: in retail and fashion industries, a strategic issue is to correctly anticipate new trends that will burst or collapse in the next weeks, months or years. To detect this subsample of trends in the fashion dataset, the following approach is proposed: using the *snaive* model, a prediction of the last year of data is computed for each trend with the associated MASE. The disrupting time series are defined as the 1000 time series where the snaive prediction has the highest MASE.

- **stable time series**: by contrast, a group of stable trends is presented. They are the 1000 time series where the *snaive* prediction with the lowest MASE.

- **seasonal time series**: we compute for each trend the seasonality strength metric introduced in Wang et al. (2006). The seasonal time series group are the 1000 fashion time series with the highest seasonality strength metric.

- **noisy time series**: Finally, several time series of the fashion dataset represent niche trends only worn and posted on social media by a few people. The average volume for these sequences is low resulting in sporadic and noisy time series difficult to forecast. In order to detect them, we use the seasonality and trend strength metrics presented in Wang et al. (2006). For each time series, the mean of these two quantities is computed and we define the time series group as the 1000 sequences with the lowest average.

Table 6 displays the MASE of each model introduced in Section 4 on the 4 subsamples of time series. Several relevant results can be highlighted. Firstly, models using the influencer weak signals strongly outperform other candidates on the disruptive time series. This result is even more visible with the HERMES approach where models using the weak signals are always better than the ones without. These results illustrate empirically the impact of weak signals on final predictions. Secondly, results of *hermes-tbats* on stable time series and seasonal time series, highlight the interest in using an hybrid model. This is even more visible on seasonal time series where hybrid approaches largely outperform neural network models presented in the benchmarks. Finally, on noisy time series, we can see that neural network models reach the highest accuracy. Furthermore, for all the HERMES approaches, we can note that the final hybrid predictions and the per-time-series predictions are the same.

## B   M4 weekly dataset, Ensembling training and results

### B.1   M4 weekly dataset

The M4 weekly dataset is a collection of 359 time series with contrasting behaviours and sizes. An overview of the dataset is given in Table 7 and some examples of sequences are given in Figure 9. This use case is not properly suited for the HERMES approach as the dataset is small and there is no clear link between time series. Moreover, no additional external signals are available that could help the RNN part to correct the first errors of the per-time-series predictors.

### B.2   M4 accuracy metrics

The M4 competition proposes 3 metrics to evaluate the different approaches: the mean absolute scaled error (MASE), the symmetric mean absolute percentage error (SMAPE) and the overall weighted average (OWA). MASE and SMAPE are defined as follow:

$$\text{MASE} = \frac{T-m}{h} \frac{\sum_{j=1}^{h} |Y_{T+j} - \hat{Y}_{T+j}|}{\sum_{i=1}^{T-m} |Y_i - Y_{i-m}|}, \qquad \text{SMAPE} = \frac{2}{h} \sum_{j=1}^{h} \frac{|Y_{T+j} - \hat{Y}_{T+j}|}{|Y_{T+j}| + |\hat{Y}_{T+j}|},$$

where h is the forecast horizon and m the length of the seasonality. The final OWA is computed by following these steps: i) compute the average MASE and SMAPE of a model. ii) Divide the previous results by the MASE and SMAPE computed with the benchmark method *snaive*. iii) Compute the OWA as the average of the relative MASE and SMAPE obtained is step ii). As an example on the M4 weekly dataset, the method *hermes-tbats* has a MASE of 7.383 and a SMAPE of 2.191. The benchmark method *snaive* has a MASE of 9.161 and a SMAPE of 2.777. Thus the OWA of *hermes-tbats* is equal to 0.797.

### B.3   FFORMA ensembling with HERMES variations

In this section, a complete description of the proposed ensembling on the M4 weekly dataset is provided. In a first time, 4 HERMES variations are trained using different per-time-series predictors. The first one

Table 6: Results summary on 4 differents subsamples of Fashion time series: i) disrupting time series, ii) stable time series, iii) seasonal time series and iv) noisy time series. Models with a * in their name have access to the external signal.

*disrupting* time series

| | MASE | |
| --- | --- | --- |
| | mean | std |
| snaive | 1.455 | - |
| thetam | 1.314 | - |
| ets | 1.27 | - |
| arima | 1.256 | - |
| tbats | 1.229 | - |
| hermes-thetam | 1.209 | 0.005 |
| hermes-ets | 1.202 | 0.007 |
| stlm | 1.198 | - |
| hermes-tbats | 1.195 | 0.01 |
| prophet | 1.193 | - |
| deepar | 1.18 | 0.03 |
| lstm | 1.15 | 0.01 |
| hermes-thetam-ws* | 1.145 | 0.019 |
| hermes-ets-ws* | 1.131 | 0.024 |
| hermes-tbats-ws* | 1.092 | 0.007 |
| lstm-ws* | 1.086 | 0.009 |

*stable* time series

| | MASE | |
| --- | --- | --- |
| | mean | std |
| prophet | 0.629 | - |
| thetam | 0.615 | - |
| ets | 0.611 | - |
| arima | 0.565 | - |
| hermes-ets-ws* | 0.555 | 0.007 |
| snaive | 0.536 | - |
| deepar | 0.531 | 0.024 |
| hermes-thetam-ws* | 0.522 | 0.004 |
| hermes-ets | 0.518 | 0.002 |
| stlm | 0.513 | - |
| hermes-thetam | 0.508 | 0.005 |
| tbats | 0.501 | - |
| lstm-ws* | 0.492 | 0.007 |
| hermes-tbats-ws* | 0.477 | 0.008 |
| lstm | 0.47 | 0.004 |
| hermes-tbats | 0.451 | 0.002 |

*seasonal* time series

| | MASE | |
| --- | --- | --- |
| | mean | std |
| snaive | 0.895 | - |
| ets | 0.895 | - |
| prophet | 0.851 | - |
| deepar | 0.836 | 0.035 |
| lstm-ws* | 0.829 | 0.014 |
| thetam | 0.826 | - |
| lstm | 0.823 | 0.013 |
| hermes-thetam | 0.815 | 0.008 |
| tbats | 0.81 | - |
| hermes-ets-ws* | 0.809 | 0.01 |
| hermes-thetam-ws* | 0.808 | 0.008 |
| arima | 0.805 | - |
| stlm | 0.786 | - |
| hermes-ets | 0.785 | 0.003 |
| hermes-tbats-ws* | 0.777 | 0.012 |
| hermes-tbats | 0.772 | 0.003 |

*noisy* time series

| | MASE | |
| --- | --- | --- |
| | mean | std |
| snaive | 0.842 | - |
| hermes-ets-ws* | 0.739 | 0.005 |
| hermes-ets | 0.726 | 0.002 |
| ets | 0.721 | - |
| thetam | 0.717 | - |
| stlm | 0.715 | - |
| prophet | 0.698 | - |
| arima | 0.696 | - |
| hermes-thetam-ws* | 0.672 | 0.003 |
| hermes-thetam | 0.669 | 0.001 |
| deepar | 0.661 | 0.007 |
| hermes-tbats-ws* | 0.647 | 0.006 |
| tbats | 0.646 | - |
| hermes-tbats | 0.644 | 0.003 |
| lstm-ws* | 0.637 | 0.004 |
| lstm | 0.636 | 0.003 |

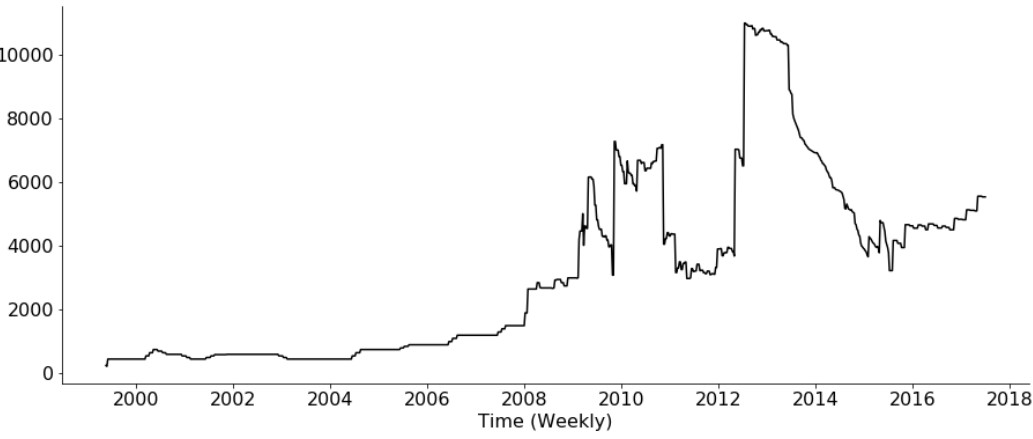

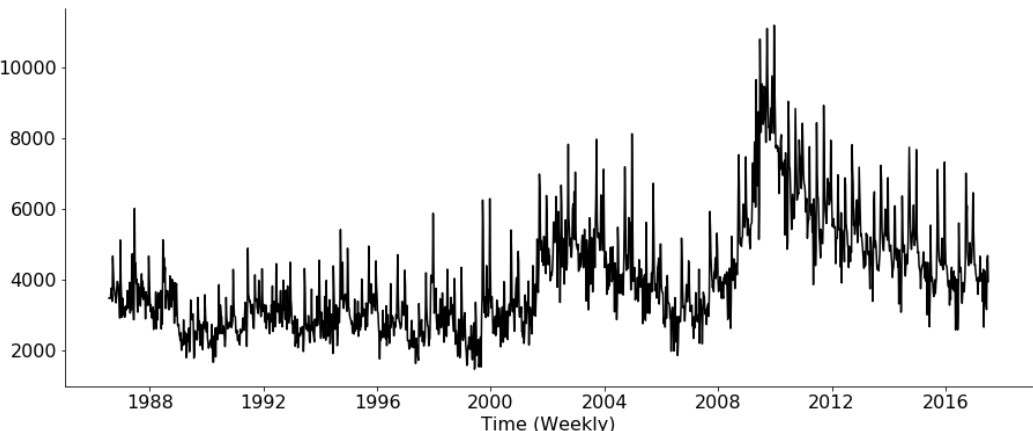

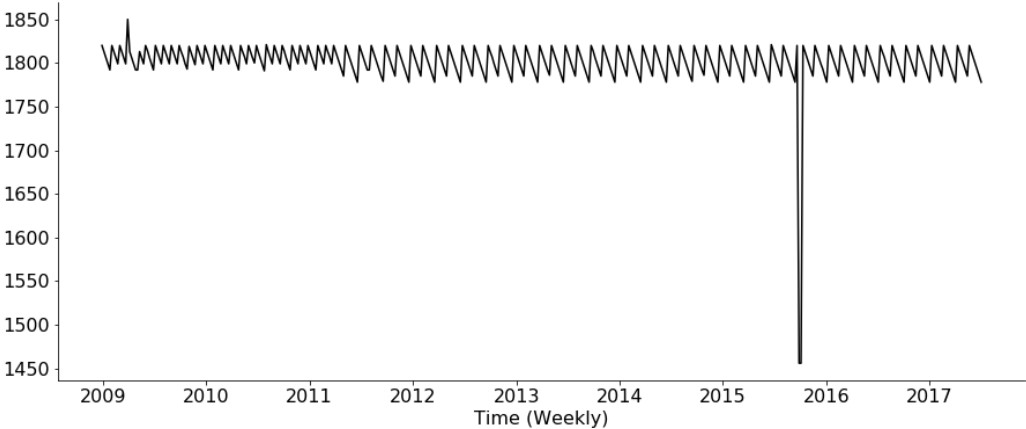

Figure 9: Examples of time series from the M4 weekly dataset. From Top to Bottom : time series called *W10* from the *Other* category, *W20* from the *Macro* category and *W220* from the *Finance* category.

Table 7: M4 weekly dataset overview. For each category, the number of sequences and the average length are given.

| | Nb. of sequences | Avg. length | Min. length |
|---|---|---|---|
| **Demographic** | 24 | 1659 | 1615 |
| **Finance** | 164 | 1237 | 260 |
| **Industry** | 6 | 834 | 356 |
| **Macro** | 41 | 1264 | 522 |
| **Micro** | 112 | 473 | **93** |
| **Other** | 12 | 1598 | 470 |

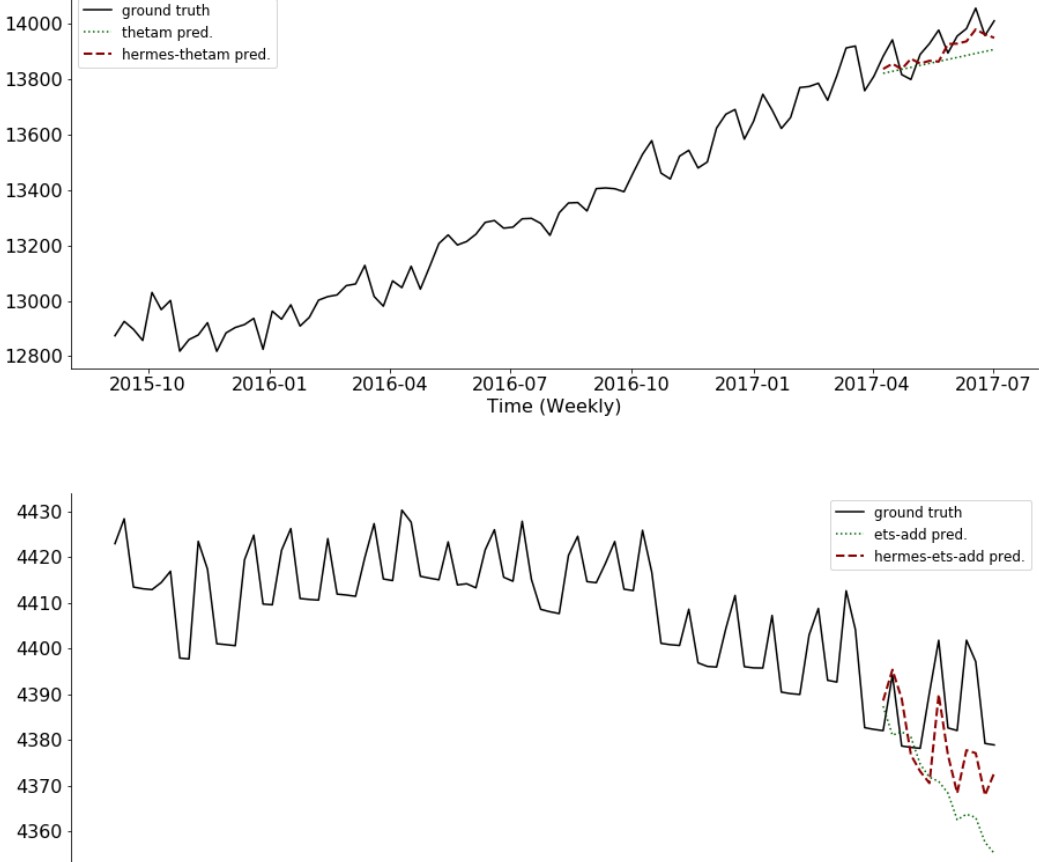

Figure 10: forecast examples of HERMES variations on 2 time series of the M4 weekly dataset. At the top, the *W133* time series is displayed with the prediction of the per-time-series predictor *thetam* (green) and the final forecast of the HERMES hybrid model *hermes-thetam* (red). At the bottom, the *W262* time series is represented with the corresponding prediction of the per-time-series predictors *ets-add* (green) and the HERMES correction of *hermes-ets-add* (red).

Table 8: Results summary on the m4 weekly dataset of the HERMES variations. For each metric, the average on all the time series is computed. For approaches using a neural network, 10 models are trained with different seeds. The mean and the standard deviation of the 10 results are displayed. For the statistical models *ets-add*, *ets-mul* and *thetam*, the Python package `statsmodels` is used. The Python package `tbats` is used for the *tbats* approach.

| | SMAPE | | MASE | | OWA | |
|---|---|---|---|---|---|---|
| | *mean* | *std* | *mean* | *std* | *mean* | *std* |
| *ets-mul* | 8.933 | - | 2.412 | - | 0.922 | - |
| *hermes-ets-mul* | 8.889 | 0.021 | 2.377 | 0.016 | 0.913 | 0.004 |
| *ets-add* | 8.929 | - | 2.410 | - | 0.921 | - |
| *hermes-ets-add* | 8.880 | 0.022 | 2.377 | 0.016 | 0.913 | 0.004 |
| *thetam* | 7.609 | - | 2.377 | - | 0.843 | - |
| *hermes-thetam* | 7.590 | 0.012 | 2.359 | 0.010 | 0.839 | 0.002 |
| *tbats* | 7.409 | - | 2.204 | - | 0.801 | - |
| **hermes-tbats** | **7.383** | 0.016 | **2.191** | 0.010 | **0.797** | 0.002 |

called *hermes-tbats* uses TBATS and is presented in Section 4.2. The second version is called *hermes-thetam* and use the Thetam method provided with the Python package `statsmodels`. The two remaining variations use as per-time-series predictors an additive or multiplicative exponential smoothing and are called respectively *hermes-ets-add* and *hermes-ets-mul*. As for Thetam, the Python package `statsmodels` is used to fit the different exponential smoothing models. Concerning the HERMES architecture, for simplicity, hyperparameters described in Section 4.2 are used for each version but a grid search could have be run for each of them. 10 models are trained per version with different seeds and the best one based on the eval set is kept for the ensemble model. In a second time, the FFORMA ensembling introduced in Montero-Manso et al. (2020) is used to combine the 4 HERMES methods. The R package `M4metalearning` containing the FFORMA model is directly used without change of the hyperparameters, imported in Python with the library `Rpy2` and combined with the HERMES code base.

### B.4 M4 weekly dataset results

In addition of the results provided is Section 4.2, Table 8 displays the results of all the HERMES variations included in the FFORMA ensembling as well as the accuracy of the per-time-series predictors. In each cases, HERMES approaches always improve the predictors accuracy. These improves can appear slight but are justified regarding the absence of link between time series and the absence of additional useful external signals. Nevertheless, efficient corrections can be obtained on some examples as displayed in Figure 10.

## C    Training parameters and loss

### C.1    Loss grid search on the fashion Dataset

Using deep learning models in time series forecasting is an appealing way to achieve higher accuracy performance. However, it induces two main issues. First, it requires a large enough dataset to train the model as illustrated in Section 4. Second, a dataset can hide contrasting time series in terms of scale, noise and behaviour. These differences can impact training performance. For the HERMES architecture, some candidate losses were defined for the training: the Mean Absolute Error (MAE), the Mean Square Error (MSE), the Scaled Mean Absolute Error (SMAE) and the Scaled Mean Square Error (SMSE). The loss functions

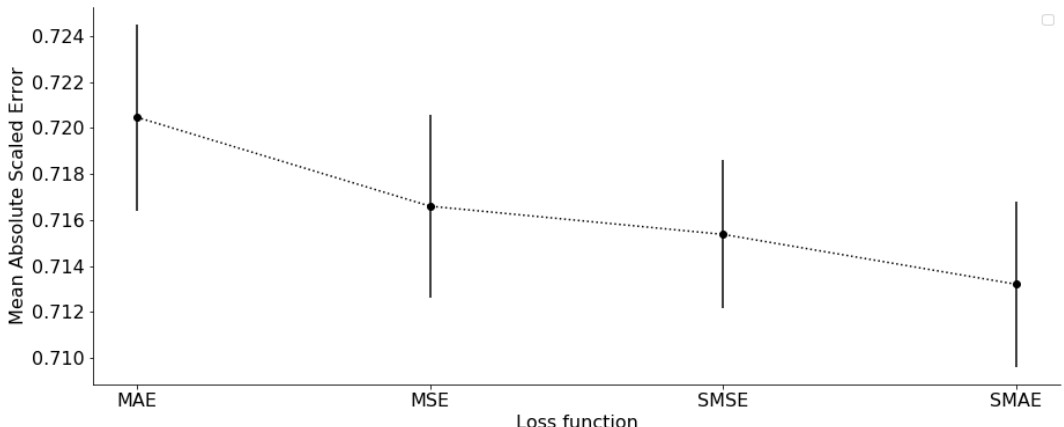

Figure 11: MASE accuray for the *hermes-tbats-ws* model depending on the loss used during the RNN training. For each loss, 10 models with different seeds have been trained. The mean and the standard deviation are represented with a point and a vertical line.

are defined as follows:

$$MAE = \frac{1}{h} \sum_{i=1}^{h} |y_{T+i}^n - \widehat{y}_{T+i|T}^n|,$$

$$MSE = \frac{1}{h} \sum_{i=1}^{h} (y_{T+i}^n - \widehat{y}_{T+i|T}^n)^2,$$

$$SMAE = \frac{1}{\overline{y}_T^n} \sum_{i=1}^{h} |y_{T+i}^n - \widehat{y}_{T+i|T}^n|,$$

$$SMSE = \frac{1}{\overline{y}_T^n} \sum_{i=1}^{h} (y_{T+i}^n - \widehat{y}_{T+i|T}^n)^2.$$

For each loss, 10 *hermes-tbats-ws* models have been trained with different seeds and the final mean and standard deviation are given in Figure 11. The final Scaled Mean Absolute Error reaches the lowest MASE and was selected to train all the HERMES models on the fashion dataset.

### C.2 Parameters grid search on the M4 weekly Dataset

In addition to the loss function, the HERMES model also depends on several hyperparameters to set correctly in order to reach satisfactory performance. For instance, an overview of the learning rate, batch size and number of windows per time series grid search for the M4 weekly dataset is shown in Figure 12. For each parameter, a collection of 10 *hermes-tbats* models have been trained with different seeds and the final OWA was calculated. As in the Figure 11, the mean and the standard deviation of each group of 10 trainings is computed. For the final *hermes-tbats* model of the M4 weekly dataset, the following set of parameters was selected: 3 windows per time series were used as the train set, the batch size was set to 8 and the learning rate was fixed to 0.005.

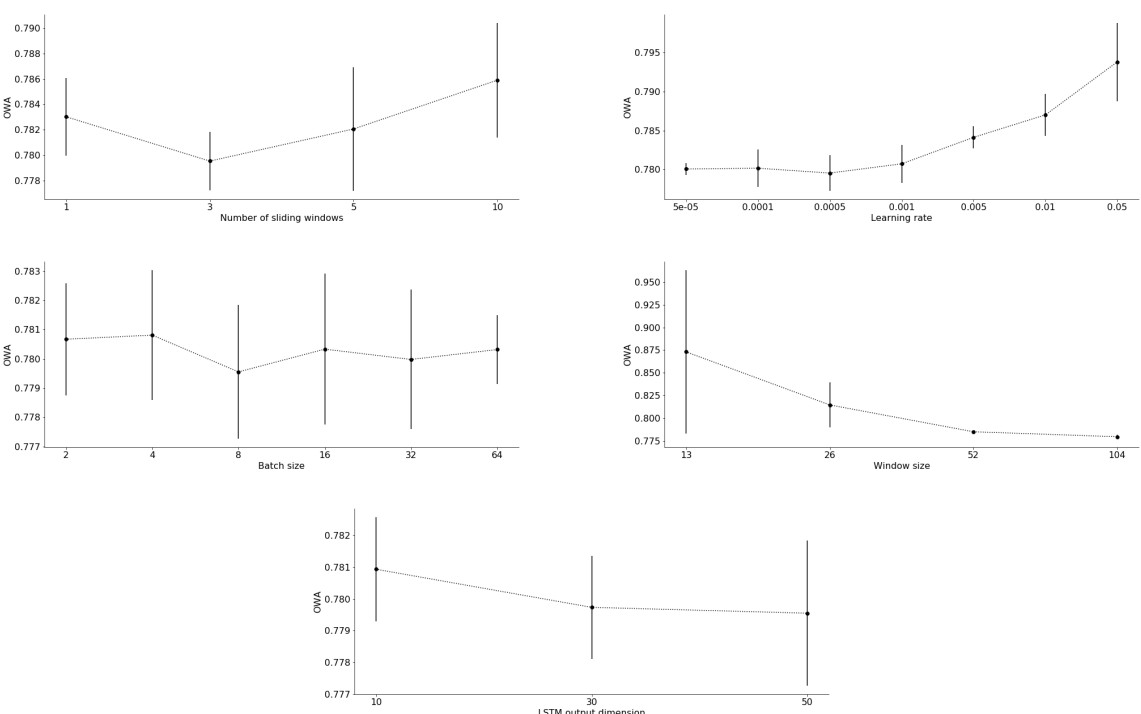

Figure 12: OWA for the *hermes-tbats* model on the eval set of the M4 weekly dataset. 5 hyperparameters used during the RNN training are tested: the number of moving windows per time series (top left), the learning rate (top right), the batch size (middle), the window size for the RNN input (bottom left) and the dimension of the LSTM layers output (bottom right). For each parameter, 10 models with different seeds have been trained. The mean and the standard deviation of the OWA on the eval set are represented with a point and a vertical line.

