# OpenReview forum: "HERMES: Hybrid Error-corrector Model with inclusion of External Signals for nonstationary fashion time series"
_TMLR — Accepted by TMLR_

### Review · Reviewer_ixuS · 2023-04-17

**Summary Of Contributions:**

This paper presents a new hybrid model for time series forecasting, called HERMES, that combines per-time-series parametric models with a global recurrent neural network (RNN) to correct the errors of the former. The paper also introduces a new fashion dataset that contains 10000 weekly fashion time series with external weak signals derived from social media influencers. The paper evaluates the performance of HERMES on the fashion dataset and the M4 weekly dataset, and compares it with several statistical and deep learning benchmarks. The paper shows that HERMES can leverage the external weak signals to improve the accuracy of forecasting and the detection of emerging and declining trends. The paper also demonstrates that HERMES is robust to different sizes of datasets and different choices of per-time-series models. The paper contributes to the literature on hybrid models for time series forecasting and provides a novel application to the fashion industry.

**Audience:**

Yes

**Broader Impact Concerns:**

1. The paper should discuss the potential positive and negative impacts of the proposed model and dataset on the society and the environment.
2. The paper should address the privacy and consent issues of using social media images and data to construct the fashion dataset and the weak signals.
3. The paper should explain how the data is collected, processed, and anonymized, and what are the ethical and legal implications of this process.
4. The paper should also acknowledge the potential risks or biases of using social media data to represent fashion trends, such as cultural diversity, inclusivity, or sustainability.


**Claims And Evidence:**

Yes

**Requested Changes:**

1. The paper should provide a clear motivation or literature review for the choice of per-time-series models.
2. The paper should explain why exponential smoothing and TBATS are chosen as the predictors, and how they compare to other possible choices such as ARIMA or HMM.
3. The paper should compare with more existing methods.
4. The paper should explain how HERMES handles different types of time series, such as stationary, non-stationary, seasonal, or sporadic.


**Strengths And Weaknesses:**

Strengths:
1. The paper proposes a novel hybrid model that can leverage external weak signals to improve the forecasting accuracy and the detection of emerging and declining trends.
2. The paper introduces a new fashion dataset that is large, diverse, and rich in information. The dataset can be useful for other researchers and practitioners in the fashion industry.
3. The paper evaluates the performance of HERMES on two datasets and compares it with several statistical and deep learning benchmarks. The paper shows that HERMES outperforms the alternatives in terms of MASE and accuracy metrics.

Weaknesses:
1. The paper does not provide a clear motivation or literature review for the choice of per-time-series models.
2. The paper could benefit from a more detailed discussion of the limitations of the proposed approach and potential avenues for future work.
3. The paper could also benefit from a more thorough comparison with existing methods, including a discussion of their strengths and weaknesses in relation to the proposed approach.

---

> ### Author Response · Authors · 2023-06-13
> **Response to review (first part)**
>
> We would like to thank the reviewer for the appreciation of the paper and the constructive feedback. We have carefully considered all comments and we will propose a revision of the paper accordingly when all reviews will be published. In the meantime, you can find below responses to some of the remarks:
>
>
> - We acknowledge that we did not discuss enough the choice of per-time-series models in the paper. This comment has led to additional justifications and discussions, in particular we discuss the use of state space models. We highlight that the main limitation for this choice is the computational time required to train the per-time-series models. In addition, in the experiment section, a third HERMES variation is added using Thetam as the per-times-series model.
>
> - Concerning the choice of per-times-series model, the main limitation is the computational time required to train these models, as the dataset contains thousands of time series. For instance, training a HMM with the EM algorithm, or training ARIMA on thousands of time series would increase significantly the computational cost. Concerning TBATS, Exponential smoothing and Thetam, correctly fitting them with existing Python packages is possible in an reasonable time, from a few minutes for exponential smoothing models to a couple of hours for TBATS. We believe that our experiment section, in particular with the additional simulations, highlights the flexibility of the hybrid approach and the numerical results motivate the use of TBATS eventhough other models could be used. To clarify this point in the paper, several justifications and comments were added in the text.
>
> - Your third remark has led to the addition of 2 new methods as benchmarks. First, the Prophet method introduced in Taylor & Letham (2017) is trained and evaluated on the fashion dataset using the Python package 'prophet'. We show that it leads to poor results in comparison to our model on the whole fashion dataset, and a comparable accuracy on the use case where only 100 time series are accessible. Secondly, the recent DeepAR method introduced in Salinas et al. (2020) is added  and trained with the Python package 'Gluonts' (Alexandrov et al., 2020). This method shows striking results on the fashion dataset and is a strong competitor of the proposed hybrid approach.
>
> Citation:
>
> Alexander Alexandrov, Konstantinos Benidis, Michael Bohlke-Schneider, Valentin Flunkert, Jan Gasthaus, Tim Januschowski, Danielle C. Maddix, Syama Rangapuram, David Salinas, Jasper Schulz, Lorenzo Stella, Ali Caner Türkmen, and Yuyang Wang. GluonTS: Probabilistic and Neural Time Series Modeling in Python. Journal of Machine Learning Research, 21(116):1–6, 2020. URL http://jmlr.org/papers/v21/
> 19-820.html.
>
> David Salinas, Valentin Flunkert, Jan Gasthaus, and Tim Januschowski. DeepAR: Probabilistic forecasting with autoregressive recurrent networks. International Journal of Forecasting, 36(3):1181–1191, 2020.
>
> Sean Taylor and Benjamin Letham. Forecasting at scale. The American Statistician, 72, 09 2017. doi:10.1080/00031305.2017.1380080.

---

### Review · Reviewer_KPmd · 2023-05-27

**Summary Of Contributions:**

This work proposes a hybrid framework for time-series forecasting. Another contribution of the work is a new time-series dataset, which tracks aggregated social media posts reflecting different fashion trends in various clothing categories and regions over a period of five years. The dataset additionally tracks social media posts of influencers across trends, categories and regions, which is to be utilised as an auxiliary signal for forecasting common-user behavior/posts. The authors fine-tune pre-trained image classifiers to setup an automated image segmentation and categorization pipeline. The pipeline is used to detect and classify fashion articles in posted images into a predefined set of categories. Images within each category are analyzed by fashion experts to identify and track various trends over time.

At the core, the hybrid forecasting framework uses a statistical forecasting model, which is optimized individually for each time-series. A recurrent neural network (RNN) is trained across all time-series to correct prediction errors of the per time-series statistical models. The authors employ available implementations of a number of statistical models in their framework.

Benchmarking results on the proposed dataset indicate that the hybrid combination of a statistical model and the error-corrector RNN outperforms other such combinations as well as individual instances of statistical and RNN models. They authors also compare variations of the RNN trained with and without the influencer activity signals to domonstrate that the best results are obtained when the signals are included as input. Experiments on smaller subsets of the fashion dataset are performed to demonstrate the applicability of the proposed model to small datasets. Finally the hybrid forecasting model and its ensemble version are shown to outperform other comparable approaches.


**Audience:**

Yes

**Broader Impact Concerns:**

The work releases a new dataset in the public domain, which is derived from social media activity of a large number of user across multiple regions. It would be in order if the authors clarify whether due to the data collection or the release of the fashion dataset, there will be any implications for individuals' privacy whose data were utilized for this work.

**Claims And Evidence:**

Yes

**Requested Changes:**


The evaluation would be more complete if performance for model trained with weak signals is reported for smaller subsets of data. Also the classification performance should be reported for smaller datasets.

The authors should also consider reporting results for multiple temporal splits of data

It seems that there is no concept of trend birth and death in the proposed dataset. It would be nice if the authors can clarify if that is true and why is that the case

**Strengths And Weaknesses:**

Strengths:

- The paper is clearly written and easy to follow

- Contribution of a new time-series dataset in fashion domain

- Competitive results on the M4 benchmark as well as on the proposed fashion dataset.

- The work evaluates a number of models


Weaknesses:

- The proposed method only provides point-forecasts. It does not provide any uncertainty bounds around predictions.

- Model optimization and evaluation only done for a single split across time (3 years train, 1 year validation, 1 test).

- Results on smaller subsets of data do not show model performance with weak signals included as input. Trend classification performance also not reported for smaller datasets.

---

### Review · Reviewer_hRzZ · 2023-06-27

**Summary Of Contributions:**

- **Constructing a new fashion time series.**
This paper presents large-scale time series data of fashion trends by processing photos posted on social networking services (SNS). The data is obtained from markets in five countries. A unique feature is the inclusion of posts by influencers tagged by domain experts.
- **Neural network models for forecasting fashion time series.** The authors propose a method that combines parametric models that capture the local behavior of time series and RNN models to include external signals as input.
- **Numerical experiments.** The effectiveness of the proposed method is evaluated using original and public data.

**Audience:**

Yes

**Claims And Evidence:**

Yes

**Requested Changes:**

- The reliability of the data is important for the use of this data by third parties. Please explain in detail how the human processing was done. Also, to what extent can machine failures be included in image processing? Do humans correct them?

- In publishing the data, it would be good to add a little more analysis of the characteristics of the data. For example, basic statistics for each market, statistics for the Weak signal $w_t^{f, i}$, etc.

- It is written that $i$ represents the index of fashion trends, but it is unclear what exactly it refers to. Does it correspond to a product/item?

- I didn't know what statistic $\hat{y}^{c,g,m}$ was. Could you please explain it in detail?

- It may be a good idea to modify the structure of Section 3.
  - It is better to explain the hybrid model (4) at first. Then, elaborate on the motivation behind the formulation of (4). The output of the RNN appears to represent the weights in summing $f^n(\cdot)$ and $\bar{y}_T^n$. Is that correct? Also, when is one of $y^{pred}$ and $y^{corr}$ more dominant than the other?
  - The cost function described in Section 4.1 should be moved to Section 3.

- It would be good to add a discussion of the limitations of this study.
  - Other external factors could be various, such as advertisements. This study focuses on the correlation between weak signals and trends, not causality. This should be clearly stated.
  - As seen in Figure 2, the effect of the external factor is accompanied by a time delay. It would be interesting to analyze how such time delays vary by fashion category and market. It would also be interesting to incorporate this into machine learning models.

- In Table 2, it would be good to clarify which methods can handle weak signals and which cannot. Then, please clarify by experiment that it is effective to construct a hybrid model by the weighted sum of two terms as in the proposed method (4).

- Minor issues:
  - The term "causal inference" appears in the abstract, but it is not discussed in this study and should be changed to another term.
  - The proposed method, HERMES, does not specify what it stands for.
  - Please cite Figure 1 in the text.
  - Please change the "t" in "At each time t" in section 2.1 to italic.

**Strengths And Weaknesses:**

S1. Creating and publishing a new fashion time series is significant because it promotes research in machine learning and data mining. The data is interesting in that it includes influencer data tagged by domain experts.

S2. Although the proposed method is somewhat incremental, it attempts to design for the characteristics of the newly created data.

S3. Experiments show that the proposed method performs better than the baseline in forecasting fashion trends.

W1. The method of data set creation (especially human processing) is unclear. Discussion of the nature and reliability of the data set is also lacking.

W2. There is room for improvement in the explanation of the proposed method.

W3. Although the authors compared the proposed method with various baselines in the experiments, there is insufficient discussion of the advantages and disadvantages of the proposed method. The results need to be organized and discussed.

---

> ### Author Response · Authors · 2023-07-10
> **Response to review (first part)**
>
> We would like to thank the reviewer for the appreciation of the paper and the constructive feedback. We have carefully considered all comments and we will propose a revision of the paper accordingly. In the meantime, you can find below detailed responses and an overview of some of the  modifications that will be added in the revised manuscript.
>
> - ``The reliability of the data is important for the use of this data by third parties. Please explain in detail how the human processing was done. Also, to what extent can machine failures be included in image processing ? Do humans correct them ?''
>
> We thank the referee for this remark. Regarding the data-acquisition pipeline presented in Section 2, the human actions are limited. The only human processing comes at the step where the fashion trends are defined. The features are automatically detected by the vision recognition models. Then, we only define trends that are relevant for the fashion industry. For instance, the Mini A-line dress fashion trend is identified where the dress category is detected with a particular shape and a particular length, all of these details detected with different models. These associations of fine-grain attributes are defined by a team of fashion experts and are not provided with the paper.
> Concerning the machine failures, all the visual recognition models used to build the proposed dataset have been trained with a standard process involving a train, a eval and a test set.
> Thus, models showing poor performance in terms of accuracy and recall on the test set have been dismissed. No additional human correction has been done. In the proposed paper, we focus on time series prediction so this has a limited impact on our claims. Of course improving visual recognition models would modify slightly the dataset but the time series  prediction challenges would remain similar.
> Please note that the data acquisition pipeline has been detailed and extended in Section 2.
>
> - ``In publishing the data, it would be good to add a little more analysis of the characteristics of the data. For example, basic statistics for each market, statistics for the Weak signal $w^{f,i}_t$, etc.''
>
> We agree with the referee and we acknowledge that we have not sufficiently presented the diversity of behaviours hidden in the proposed fashion dataset. So as to improve this point, a new section is added in the appendix of the revised version of the paper where 4 different sub-samples of time series are defined and studied separately.
>
> - ``It is written that represents the index of fashion trends, but it is unclear what exactly it refers to. Does it correspond to a product/item?''
>
> We understand this remark and further information has been added in the revised paper to clarify this point. The index of fashion trends represents the name of a fashion clothing relevant for the fashion industry. It can be just one attribute detected by a visual recognition model (ex: the Sneakers) but it can also be a combination of several attributes (ex: the Slim Sole Retro Basketball Sneakers). As the fashion trends names are anonymized in the proposed dataset, the architecture used to define the fashion trends is not presented in this paper.
>
> - ``I didn't know what statistic was $\hat{y}^{c,g,m}$. Could you please explain it in detail?''
>
> We are not sure of the notation in the question of the referee, in the paper we introduce $\bar{y}^{c,g,m}$ and $\tilde{y}^{c,g,m}$ but not $\hat{y}^{c,g,m}$. We present below a explanation of the two statistic $\bar{y}^{c,g,m}$ and $\tilde{y}^{c,g,m}$:
>
> . $\tilde{y}^{c,g,m}$ is the time series representing the behaviour of the global cloth type $c$ for the gender $g$ on the market $m$. It can be for example the time series representing the evolution of the general pants for females in Europe. The sequence $\tilde{y}^{c,g,m}$ is then used to normalize each time series $y^{c,g,m,j}$ with the same $c$, $g$ and $m$. All the fashion trends of pants for females in Europe are normalized by the time series representing the general pants for females in Europe.
>
> . $\bar{y}^{c,g,m}$ is the time series $\tilde{y}^{c,g,m}$ where the seasonality component has been removed. The seasonality component is removed as we only want to remove social media bias with the normalization step and not remove or change the seasonality of the proposed fashion trends.
>
> Additional information (regarding the Jersey Top normalization example) has been added at the end of Section 2.2 and in the legend to Figure 3.

---

> > ### Author Response · Authors · 2023-07-10
> > **Response to review (second part)**
> >
> > - ``It may be a good idea to modify the structure of Section 3. It is better to explain the hybrid model (4) at first.  Then, elaborate on the motivation behind the formulation of (4). The output of the RNN appears to represent the weights in summing $f^n(.)$ and $\bar{y}^n_{T}$. Is that correct? Also, when is one of $y^{pred}$ and $y^{corr}$ more dominant than the other? The cost function described in Section 4.1 should be moved to Section 3.''
> >
> > We thank the referee for this remark. In the proposed method, $f^n(.)$ represents a time-series-specific parametric model, the model $f^n(y^n_{1:T},\theta^n_{predictor})$ is linked to the time series $y^n_{1:T}$. By contrast, the recurrent neural network denoted $RNN()$ in the paper is global: the same network is used for every time series. It is why we introduce $\bar{y}^n_T$ to rescale inputs and outputs of $RNN()$ as we do not assume that all the time series have the same volume. The fraction of the final prediction due to $y^{pred}$ or $y^{corr}$ is variable and depends on the nature of the time series (and the external signal if it is provided to the corrector model). An illustration is presented in a new section added in the appendix of the revised paper. In this section, we study the accuracy of the HERMES model on sub-samples of time series sharing the same behaviour. On the sub-sample of time series with a high level of noise, predictions of the HERMES model seem to rely only on $y^{pred}$. By contrast, on a sub-sample called 'disrupting trend', we can see that $y^{corr}$ strongly impacts the final prediction resulting in an improvement of the final accuracy of the HERMES model. Concerning the cost function described in Section 4.1, we decided to present this part in the Experiments section and not in the section presenting the model as the choice of the loss function can change depending on the use case. For instance, the L1 loss function was selected to train the HERMES variations on the Fashion dataset while the MASE loss has been used for the M4 dataset.
> >
> > - ``It would be good to add a discussion of the limitations of this study. Other external factors could be various, such as advertisements. This study focuses on the correlation between weak signals and trends, not causality. This should be clearly stated. As seen in Figure 2, the effect of the external factor is accompanied by a time delay. It would be interesting to analyze how such time delays vary by fashion category and market. It would also be interesting to incorporate this into machine learning models.''
> >
> >  We thank the referee for this remark and supplementary information has been added in the result sections and in the appendix of the revised paper. As future works on the dataset, we attempt to build a second fashion dataset gathering more time series and additional external signals such as fashion shows signals, brands signals etc...
> > Concerning the forecasting model, we acknowledge that we did not discuss enough about the limits of the proposed approach. In the revised version of the paper, a discussion section is added at the end of Section 4. For instance, we agree that designing models to explain the time delay between weak signals shifts and their consequences on the target signals is an interesting objective. We believe that this is out of the scope of the present paper but this is the topic of an ongoing work using hidden states to leverage  the impact of external signals on the main signal.
> >
> > - ``In Table 2, it would be good to clarify which methods can handle weak signals and which cannot. Then, please clarify by experiment that it is effective to construct a hybrid model by the weighted sum of two terms as in the proposed method (4).''
> >
> > In the revised version of the paper, a star is added in Table 2 and Table 4 to specify which methods have access to the weak signals and which do not. In addition, in Table 4, results of HERMES declinations with and without the weak signals are added. Finally, so as to provide more benchmarks, the Prophet method, the DeepAR model and a third HERMES variation are added for the Fashion use case.
> >
> > - ``Minor issues: i)The term "causal inference" appears in the abstract, but it is not discussed in this study and should be changed to another term. ii) The proposed method, HERMES, does not specify what it stands for. iii) Please cite Figure 1 in the text. iv) Please change the "t" in "At each time t" in section 2.1 to italic.''
> >
> > We thank the referee for these remarks and the text has been corrected accordingly. Concerning the remark on the name of the proposed model, HERMES stands of Hybrid ERror-corrector Model with inclusion of External Signals.

---

### Review · Reviewer_o2ju · 2023-07-09

**Summary Of Contributions:**

This paper makes two contributions. First, it provides a large fashion time series generated from a massive collection of images. Second, it proposes a forecasting model in the specific setting of fashion trend prediction. Each fashion trend is basically modeled independently as a univariate signal, but the model is designed to incorporate "weak signals," which may result in some dependency between the univariate time series.

**Audience:**

Yes

**Broader Impact Concerns:**

Dataset is valuable and the paper makes a good contribution to the community.

**Claims And Evidence:**

Yes

**Requested Changes:**

Improve the description of Section 2.
- Clearly define the terms like trend and cloth type using examples.
- Section 2 should be described in a top-down manner if the resulting dataset is claimed to be one of the main contributions. Describe first what the resulting data looks like, then explain how you generated it.
- It is not clear if the authors are calming technical contributions in the preprocess approaches. If so, the authors need to validate the specific choices like (1). The figure looks like a cherry-picked example.


Section 3
- The univariate model adopted is not clearly described. What is the model?
- The term "error correction" sounds misleading as there is not such an explicit algorithm to compute the error. Perhaps the second term of (3) should be called just the correction term.
- I need methodological validation on (4). Apparently, the scaling of the input of RNN is critical to serving the RNN term as the coefficient of the correction term.


**Strengths And Weaknesses:**

Strength
- Dataset is very interesting and should be valuable to the community.
- Introduces many fashion-specific heuristic preprocesses, based on domain knowledge.

Weakness
- The description is often hard to follow possibly due to language issues.
- The novelty of the prediction model looks weak.

---

> ### Author Response · Authors · 2023-07-12
> **Response to review**
>
>
> We would like to thank the reviewer for the appreciation of the paper and the constructive feedback. We have carefully considered all comments and we will propose a revision of the paper accordingly. In the meantime, you can find below detailed responses and an overview of some of the  modifications that will be added in the revised manuscript.
>
> - Clearly define the terms like trend and cloth type using examples.
>
> We thank the referee for this remark and we acknowledge that we have not presented enough the definition of what we call a fashion trend. In the revision of the article, additional information has been added to clarify this point as well as a complete example of the Jersey top trend for women in China. We also provided additional information on the dataset and on the fashion time series in Section 2 of the revised paper.
>
> - Section 2 should be described in a top-down manner if the resulting dataset is claimed to be one of the main contributions. Describe first what the resulting data looks like, then explain how you generated it
>
> We understand the remark of the reviewer and reversing the order could be a nicer way to present the fashion dataset. However, regarding the reviewing timeline, we will not be able to deeply change the structure of Section 2 before the end of the process. We apologize for that and we hope that the current structure of Section 2 and all the modifications made in the revised version are  acceptable.
>
> - It is not clear if the authors are calming technical contributions in the preprocess approaches. If so, the authors need to validate the specific choices like (1). The figure looks like a cherry-picked example.
>
> We thank the referee for this remark. The preprocess approach presented in (1) is not presented as a technical contribution but  a  way to normalize data from social networks. We gave a few details to explain this normalization in order to explain how the dataset provided with the paper was produced. In Section 2, we explain the process to obtain time series from social media data. These steps are not a contribution as we only used standard signal processing techniques. However, we believe that they help understanding the dataset which is a first contribution of the paper as it is proposed publicly. The presentation of the dataset has been modified in Section 2 and in Appendix A, we hope that this clarifies the contribution of the paper.
>
> - The univariate model adopted is not clearly described. What is the model?
>
> We thank the referee for this remark. One of the novelties of the proposed approach is that we present a general hybrid framework able to include all univariate models and it is why we only introduced a general notation $f^n()$ in Section 3 for this model. In the paper we propose different models: exponential smoothing, Thetam and TBATS. These models are given with the references as they are not the contribution of the paper, but could be added in the appendix if necessary.
> We acknowledge that this feature of the proposed model was not sufficiently highlighted in the original paper. Consequently, additional information has been added in Section 3 and a third HERMES variation has been added in Section 4 to illustrate this point.
>
> - The term "error correction" sounds misleading as there is not such an explicit algorithm to compute the error. Perhaps the second term of (3) should be called just the correction term.
>
> We thank the referee for these remarks and the text has been corrected accordingly.
>
> - I need methodological validation on (4). Apparently, the scaling of the input of RNN is critical to serving the RNN term as the coefficient of the correction term.
>
> The 10000 time series in the fashion dataset have different volumes, from $10^{-5}$ for niche trends to $10^{-1}$ for common trends. As we want to train the RNN part on the whole dataset to learn cross dynamics, the normalization step of the RNN input is inevitable. Two methods have been tested and are discussed at the end of Section 3.2. i) The RNN input can be normalized by directly dividing it by the prediction of the univariate model prediction. ii) The RNN input can be normalized by the mean volume of the time series. The first solution presents a main issue that it shows instabilities with time series close to 0 and it can simply not be used if the prediction of the univariate model prediction hits 0. Consequently, the second solution is used in the proposed hybrid framework.
> Of course, other scaling procedures could be used. In this paper this is not the main topic of interest and this step is used to design a training procedure that can deal with thousands of time series. Analyzing in detail the impact of the scaling procedure on the overall training is left for future works.

---

> > ### Comment · Reviewer_o2ju · 2023-07-12
> > **Thanks for the clarifications**
> >
> > I understand the review was delayed. My comment is based on the original submission. If the data preprocessing is a sort of heuristic chosen in an ad hoc manner (that's perfectly fine, if you are not claiming technical contributions there), you may want to mention it clearly to avoid confusion.

---

> > > ### Author Response · Authors · 2023-07-22
> > > **Response to review**
> > >
> > > We agree with the referee and the text has been corrected accordingly.

---

### Decision · Action_Editors · 2023-08-15

**Recommendation:** Accept with minor revision

**Comment:**

This paper proposes a new fashion time-series data set and proposes a RNN-based time-series forecasting method. Specifically, it first offers an extensive fashion time series dataset generated from an extensive collection of images. Secondly, it introduces a forecasting model tailored to the realm of predicting fashion trends. While each fashion trend is treated as an independent univariate signal, the model is constructed to accommodate "weak signals," potentially introducing interdependencies among these individual time series.

Overall, the paper is well written. The issues raised by the reviewers have been adequately resolved. Therefore, I also recommend accepting it.

As a minor change, the size of the figures in Figure 12 is not consistent. Please make it consistent in the final version.

**Audience:**

This paper proposes an extensive fashion time series dataset. This will attract not only fashion time-series modeling researchers but also machine learning researchers who are working on time-series modeling. Thus, this paper will attract many researchers in the community.

**Claims And Evidence:**

Yes.

---

> ### Comment · Action_Editors · 2023-09-06
> **Please modify the manuscript**
>
> Dear authors,
>
> It seems the authors are using the non-camera-ready format. The header of the manuscript is still "Under review as submission to TMLR". Please modify it as "Published in Transactions on Machine Learning Research (09/2023)".
>
> So, please update it.
>
> Thanks,
>
> AC